



# Colorado air quality impacted by long range transport: A set of case studies during the 2015 Pacific Northwest fires

Jessie M. Creamean[1,2*], Paul. J. Neiman[2], Timothy Coleman[1,2], Christoph J. Senff[1,3], Guillaume Kirgis[1,3], Raul J. Alvarez[3], and Atsushi Yamamoto[4]

[1]University of Colorado at Boulder, Cooperative Institute for Research in Environmental Sciences, Boulder, CO, 80309, USA
[2]NOAA Earth System Research Laboratory, Physical Sciences Division, Boulder, CO, 80305, USA
[3]NOAA Earth System Research Laboratory, Chemical Sciences Division, Boulder, CO, 80305, USA
[4]Process and Environmental, HORIBA, Instruments, Inc., Irvine, CA, 92618, USA

*Correspondence to*: Jessie M. Creamean (jessie.creamean@noaa.gov)

**Abstract.** Biomass burning plumes containing aerosols from forest fires can be transported long distances, which can ultimately impact climate and air quality in regions far from the source. Interestingly, these fires can inject aerosols other than smoke into the atmosphere, which very few studies have evidenced. Here, we demonstrate a set of case studies of long-range transport of mineral dust aerosols in addition to smoke from numerous forest fires in the Pacific Northwest to Colorado, U.S. These aerosols were detected in Boulder, Colorado along the Front Range using Beta-ray attenuation and energy dispersive X-ray fluorescence spectroscopy, and corroborated with satellite-borne lidar observations of smoke and dust. Further, we examined the transport pathways of these aerosols using air mass trajectory analysis and regional and synoptic scale meteorological dynamics. Three separate events with poor air quality and increased mass concentrations of metals from biomass burning (S and K) and minerals (Al, Si, Ca, Fe, and Ti) occurred due to the introduction of smoke and dust from regional and synoptic scale winds. Cleaner time periods with good air quality and lesser concentrations of biomass burning and mineral metals between the haze events were due to the advection of smoke and dust away from the region. Dust and smoke present in biomass burning haze can have diverse impacts on visibility, health, cloud formation, and surface radiation. Thus, it is important to understand how aerosol populations can be influenced by long-range transported aerosols, particularly those emitted from large source contributors such as forest fires.

**Keywords.** Aerosol transport, air quality, mineral dust, biomass burning, remote sensing, in situ observations

## 1 Introduction

Wildfires in both forested and agricultural regions serve as a steady source of pollutants into the atmosphere. Gas phase constituents such as methane ($CH_4$), carbon monoxide (CO), carbon dioxide ($CO_2$), sulphur dioxide ($SO_2$) and nitrogen oxides ($NO_x$; $NO + NO_2$) can be produced from burning of biofuels (Gadi et al., 2003; Radojevic, 2003), in addition to precursors that induce ozone production (Jaffe and Wigder, 2012). Additionally, wildfires produce large concentrations of aerosols which are injected into the atmosphere or formed in the smoke plume via secondary processes and include carbonaceous species (elemental and organic carbon) (Park et al., 2003; Spracklen et al., 2007)





and biogenic heavy metals (including but not limited to Fe, Mn, Cd, Cu, Pb, Cr, and Ni) (Nriagu, 1989; Radojevic,
2003). Soluble inorganic species such as sulphate, nitrate, ammonium, and chloride are found in fire emissions and
partitioned to the particle phase through heterogeneous reactions with the gas phase species released during the
combustion process (Pio et al., 2008). Strong, turbulent winds inside combustion zones from controlled and wild
vegetation fires can introduce considerable amounts of dust particles into the free troposphere, which can subsequently
be transported over thousands of kilometres with the smoke (Clements et al., 2008; Ansmann et al., 2009; Baars et al.,
2011). However, few studies have documented how wildfires inject mineral dust into the atmosphere (Gaudichet et
al., 1995; Chalbot et al., 2013; Yang et al., 2013; Nisantzi et al., 2014), particularly in heavily forested or agricultural
regions such as the Pacific Northwest of the U.S. where dust sources are limited relative to arid regions in Africa, the
Middle East, and Asia. Prescribed burning (i.e., slash-and-burn techniques) and wildfires are common in these arid
"dust belt" regions, inducing the simultaneous emission of dust and smoke (Streets et al., 2003; Pinker et al., 2010).

Aerosols produced directly from wildfires (i.e., carbonaceous and soluble inorganic particulates) or injected into the
free troposphere from smoke plume dynamics (i.e., mineral dust) have diverse effects on climate and air quality. For
instance, hygroscopic organic aerosol, sulphate, and nitrate can enable aerosols to serve as cloud condensation nuclei
(CCN) (Cruz and Pandis, 1997), whereas mineral dust and black carbon are effective ice nucleating particles (INPs)
at sub-freezing temperatures (DeMott et al., 1999; DeMott et al., 2003; Vali et al., 2015). Both of these aerosol nuclei
modify cloud radiative properties, lifetime, and impact precipitation formation, and have been shown to originate from
prescribed burns and wildfires (Eagan et al., 1974; McCluskey et al., 2014). Enhanced pollutants from fires also
severely influence air quality, and can prompt adverse health effects (Bravo et al., 2002; Phuleria et al., 2005;
Wiedinmyer et al., 2006). For instance, smoke plumes from wildfires have been linked to childhood mortality
(Jayachandran, 2008), asthma (Bowman and Johnston, 2005), and various respiratory illness and diseases (Mott et al.,
2002; Moore et al., 2006). These effects are additionally complicated by aging from biogenic gases in the smoke
plume during transport. Further, previous air quality studies on the East Coast of the U.S. have shown that enhanced
aerosol optical depths associated with both wildfires and anthropogenic sources can cause large errors in
meteorological models used to forecast poor air quality events (Zamora et al., 2005). Overall, the aerosol species
emitted or formed from wildfire plumes are complex in nature and possess several diverse climate and health effects,
thus demonstrating the need to better understand the various types, sources, and transport pathways of these emissions.

Air quality is strongly dependent not only on emission sources such as wildfires, but also on weather and climate
change (Jacob and Winner, 2009). Regions with complex topography such as the Front Range of Colorado, U.S. (see
Figure 1) have unique meteorological phenomena such as upslope/downslope flows that serve as agents for focusing
or cleaning out local air pollution from the Denver metropolitan area (Haagenson, 1979). Typically, this region is
characterized by good air quality in terms of particulate matter (PM) relative to other larger urban and industrial areas,
although it experiences occasional pollution episodes due to modulation of the mountain slope dynamics, oil and
natural gas production, and wildfires (Watson et al., 1998; Sibold and Veblen, 2006; Brown et al., 2013). Here, we
show that the Front Range air quality was severely impacted by long-range transported wildfire emissions from the



Pacific Northwest during Aug 2015. A reoccurring influx of pollutants, including $SO_2$, $NO_x$, and smoke aerosols,
infiltrated the Front Range region due to shifts in regional and synoptic scale meteorology. Interestingly, mineral dust
was also transported with the smoke plume to the Front Range from the wildfires. This complex mixture of gases and
aerosols can have numerous climate and health effects in the region, and should be evaluated to develop a better
understanding of future influences from wildfire emissions, especially considering a warmer and drier climate will
potentially lead to more frequent wildfires (Westerling et al., 2006; Liu et al., 2010).
**2 Methods**
**2.1 Satellite observations**
The source of aerosols from the fires was determined using imagery from the Moderate Resolution Imaging
Spectroradiometer (MODIS) on board the Terra satellite. MODIS is a multi-spectral sensor with 36 spectral bands,
ranging in wavelength from 0.4 to 14.2 µm. Aerosol optical depth (AOD) data from MODIS were acquired from the
Giovanni data server (http://giovanni.gsfc.nasa.gov/giovanni/) for daily AOD at a 1° spatial resolution using a domain
of 82°W to 163°W and 26°N to 59°N (MOD08_D3_051). MODIS AOD is retrieved from three spectral channels
(0.47 µm, 0.66 µm, and 2.1 µm) using the algorithm described by Kaufman et al. (1997) in cloud-free pixels (10 km
x 10 km grid box) (Ackerman et al., 1998). Fire and surface thermal anomaly data were also acquired from the MODIS
Terra    satellite    using    brightness    temperature    measurements    in    the    4-µm    and    11-µm    channels
(https://earthdata.nasa.gov/labs/worldview/) (Giglio, 2010). The fire detection strategy is based on absolute detection
of a fire (when the fire strength is sufficient to detect), and on detection relative to its background (to account for
variability of the surface temperature and reflection by sunlight) (Giglio et al., 2003). The algorithms include masking
of clouds, bright surfaces, glint, and other potential false alarms (Giglio et al., 2003). Swaths from overpasses over
the Pacific Northwest were used to determine the locations of fires on a daily basis.

In order to evaluate the types of aerosols present in enhanced AOD plumes over the western U.S., aerosol subtype
data were retrieved from Cloud-Aerosol Lidar with Orthogonal Polarization (CALIOP) on board Cloud-Aerosol Lidar
and Infrared Pathfinder Satellite Observations (CALIPSO). Level-2 ValStage1 V.30 Vertical Feature Mask data
obtained    from    NASA's    Earth    Observing    System    Data    and    Information    System    (EOSDIS;
https://search.earthdata.nasa.gov/) contain vertically-resolved data of aerosol layer sub-type, including but not limited
to smoke, dust, and polluted dust (i.e., dust mixed with smoke) (Vaughan et al., 2004; Omar et al., 2009; Winker et
al., 2009). CALIPSO was launched on 28 Apr 2006 and flies in an orbital altitude of 705 km as part of the sun-
synchronous "A-train" satellite constellation. CALIOP is an elastic backscatter lidar operating at 532 nm and 1064
nm, completed with a depolarization channel at 532 nm to enable detection of aerosols and clouds. Granule data were
acquired from orbital swaths that passed over the north-western U.S. (domain includes Washington, Oregon, northern
California, Idaho, Nevada, Montana, Wyoming, Utah, and Colorado) from 15 Aug to 2 Sep 2015 and processed using
modified Python code developed by the Hierarchical Data Format (HDF) group at the University of Illinois, Urbana-
Champaign (http://hdfeos.org/). Aerosol sub-types were also examined off the U.S. West Coast across the central





North Pacific Ocean, in the context of air mass trajectory analysis, to ensure mineral dust and smoke were transported
to Colorado from the Pacific Northwest fires rather than from deserts or fires overseas.

**2.2 Colorado air quality data**

All air quality data were acquired from the Colorado Department of Public Health and Environment (CDPHE;
http://www.colorado.gov/airquality/report.aspx) from 15 Aug to 2 Sep 2015 at various sites throughout the Colorado
Front Range (see Figure 1). Table 1 provides the site latitudes, longitudes, elevations, and which measurements were
available at each site. Hourly measurements included mass concentrations ($\mu$g m$^{-3}$) of particulate matter for particles
with diameters $\leq 2.5$ $\mu$m ($PM_{2.5}$) and $\leq 10$ $\mu$m ($PM_{10}$). Carbon monoxide (CO), sulphur dioxide ($SO_2$), nitric oxide
(NO), nitrous oxide ($NO_2$), and ozone ($O_3$) were also evaluated but no significant differences were observed between
haze and non-haze time periods, thus the data are not presented. All times shown are coordinated universal time [UTC;
local time or mountain daylight time (MDT) + 6].

**2.3 In situ aerosol observations at Boulder, Colorado**

Real-time, hourly ambient aerosols samples were analysed for $PM_{2.5}$ total mass concentrations ($\mu$g m$^{-3}$) and
concentrations of various metals (ng m$^{-3}$) using the HORIBA, Ltd. PX-375 continuous particle mass and elemental
speciation monitor (http://www.horiba.com/process-environmental/products/ambient/details/continuous-particulate-
monitor-with-x-ray-fluorescence-px-375-27871/) from 26 Aug to 2 Sep 2015 at the National Oceanic and
Atmospheric Administration (NOAA) David Skaggs Research Centre (DSRC) located in Boulder, Colorado (39.99°N,
105.26°W, and 1672 m MSL; see Figure 1). The PX-375 draws in air at 16.7 L min$^{-1}$ through a U.S. Environmental
Protection Agency (EPA) Louvered $PM_{10}$ inlet, then subsequently passes through a BGI Very Sharp Cut Cyclone
(VSCC™) to filter for particles smaller than 2.5 $\mu$m in diameter. Air is pulled through a nozzle for 60 minutes per
hourly sample, where particles are subsequently deposited in a 100-mm diameter spot on Teflon™ PTFE fabric filter
tape for analysis. Once the sample is collected for 60 minutes, beta-ray attenuation and energy dispersive X-ray
fluorescence spectroscopy (EDXRF) analyses are conducted for 60 minutes and 1000 seconds, respectively, per hourly
sample, simultaneous to the sampling of the subsequent sample. Beta-ray attenuation analysis is used to measure total
$PM_{2.5}$ mass concentrations and EDXRF is used to analyse concentrations of Ti, V, Cr, Mn, Fe, Ni, Cu, Zn, As, Pb, Al,
Si, S, K, and Ca. The EDXRF unit contains a CMOS camera for sample images. Calibration material used for X-ray
intensity is NIST SRM 2783. Lower detection limits (LDLs) are shown in Table 2 and error was calculated to be $\pm 2\%$
for hourly metal concentrations. Hourly total $PM_{2.5}$ mass concentrations had an LDL of 2.00 g $\mu$m$^{-3}$.

**2.4 Aerosol and ozone remote sensing observations at Boulder, Colorado**

The Tunable Optical Profiler for Aerosol and oZone (TOPAZ) lidar was operated at the DSRC on 9 days from 14 Aug
through 2 Sep 2015 and it collected about 62 hours of ozone and aerosol profile data, primarily between mid-morning
and early evening local time. TOPAZ is a state-of-the-art, tunable ozone differential absorption lidar. It emits pulsed
laser light at three ultraviolet wavelengths between 285 and 295 nm and measures ozone as well as aerosol backscatter
and extinction profiles with high temporal and spatial resolutions (Alvarez et al., 2011). The TOPAZ lidar is mounted




in a truck with a rooftop two-axis scanner. This scanner permits pointing the lidar beam at elevation angles between
−5 and 30 degrees at a fixed but changeable azimuth angle. To achieve zenith operation the scanner mirror is moved
out of the beam path. Typical TOPAZ operation consists of a scan sequence at 2, 6, 20, and 90 degrees elevation,
repeated approximately every five minutes. The range-resolved ozone and aerosol observations at the shallow
elevations angles are projected onto the vertical and spliced together with the zenith observations, resulting in
composite vertical ozone and aerosol profiles from about 15 m to 2–3 km above ground level (AGL) at five minutes
time resolution (Alvarez et al., 2012). In this study, we only used the lidar aerosol extinction profiles measured at a
wavelength of 294 nm. The aerosol profile retrieval requires assumptions about the lidar calibration constant and the
aerosol extinction-to-backscatter or lidar ratio. For this study we used an altitude-constant lidar ratio of 40 sr, which
is a good approximation for continental and urban aerosols. The lidar signal at the aerosol wavelength of 294 nm is
also affected by ozone absorption. Therefore, uncertainties in the ozone observations can cause biases in the aerosol
retrieval. This, combined with uncertainties in the calibration constant and lidar ratio, can lead to errors in the aerosol
extinction coefficient profiles of up to about 30%. The precision of the 5-minute aerosol extinction measurements is
typically better than 10%.
**2.5 Meteorological data and analysis**
A gridded perspective of synoptic-scale conditions across North America was provided using the NOAA/National
Centres for Environmental Prediction (NCEP) Rapid Refresh numerical data package [RAP;
http://rapidrefresh.noaa.gov/ (Benjamin et al., 2016)]. The RAP is an operational assimilation/modelling system
updated hourly, with 13-km horizontal resolution and 50 vertical levels.

Air mass backward trajectory analyses were conducted using HYSPLIT 4 (Draxler and Rolph, 2011) and data from
the NOAA/NCEP Global Data Assimilation System (GDAS) (Kalnay et al., 1996). HYSPLIT trajectories do not
include processes that may affect particle concentrations such as convective transport, wet removal, or dry removal,
and are only intended to highlight the possible transport pathways. To study the potential for transport from the Pacific
Northwest fires region, and to eliminate potential contribution from aerosol sources overseas, we used an ensemble of
backward trajectories initiated at multiple altitudes and times ending above the NOAA building in Boulder. Ten-day
back trajectories were initiated every 6 hours (at 00:00, 06:00, 12:00, and 18:00 UTC) during 15 Aug–2 Sep 2015 at
500, 1000, and 2000 m AGL (corresponds to 2172, 2672, and 3672 m MSL).

A 449-MHz wind profiler (White et al., 2013), deployed near the Boulder Atmospheric Observatory in Erie, Colorado
(BAO; 40.05N, 105.01°W, and 1577 m MSL; location shown in Figure 1), provided hourly-averaged profiles of
horizontal wind. The high (low) mode extended from 145 m (195 m) to 10074 m (5059 m) AGL with a vertical
resolution of 200 m (100 m). The wind-profiler data were edited objectively using the vertical-temporal continuity
method of Weber et al. (1993) and then subjected to additional manual editing as needed. For the purpose of this study,
we utilized only the low-mode observations.



## 3 Results and discussion

### 3.1 Haze events induced poor air quality along Colorado's Front Range

The shift in air quality was evident during three August haze events in the Denver metro area. Figure 2 shows photos of notable air quality transitions in Denver looking westward towards the foothills of the Rocky Mountains. The image on 15 Aug shows typical, clean conditions, where the foothills were visible west of Denver. On 17 Aug, a haze settled in the region, creating a low-level pollution plume that masked the view of the foothills. This haze continued to infiltrate the Denver metro area, reaching the poorest visibility on 23 Aug. This haze persisted in the Denver metro area until 27 Aug, when clear conditions were re-established and the foothills were once again visible. However, the air quality deteriorated once again by 29 Aug, with hazy conditions obscuring the foothills. This haze event was shorter lived, clearing out once again on 31 Aug. The cleaner conditions persisted until the end of the measurement period on 2 Sep. The qualitative observations of the three separate haze events were corroborated by in situ air quality measurements along the Front Range. Figure 3 shows hourly and daily averaged $PM_{2.5}$ mass concentrations (herein, simply called "$PM_{2.5}$") at the sites provided in Table 1. Overall, three separate haze events occurred along the Front Range with the worst days visually observed (Figure 2) on the 17, 23, and 29 Aug (events 1, 2, and 3, respectively), when $PM_{2.5}$ reached maximum concentrations and a cold front passed through (discussed in section 3.3). Prior to each of these events, $PM_{2.5}$ was suppressed then slowly increased to each event's maximum concentrations on 17, 23, and 29 Aug. $PM_{2.5}$ slowly decreased following each of these haze events. $PM_{10}$ (not shown) did not follow similar increases and decreases as the $PM_{2.5}$, suggesting the smaller particles contributing to $PM_{2.5}$ originated from different, likely more distant sources as compared to coarser particles contributing to the $PM_{10}$, which are likely from more local sources (VanCuren, 2003; Neff et al., 2008).

### 3.2 Biomass burning plume propagates towards Colorado

During the 15 Aug–2 Sep time period, fires in both forested and agricultural vegetation areas and to some extent in shrub and grasslands in the Pacific Northwest were prominent. Figures 4–6 show MODIS retrievals of fire hotspots and aerosol optical depth (AOD) during the first, second, and third haze events in Colorado, when numerous fires were detected in Washington, Oregon, northern California, northern Idaho, and north-western Montana. Three cases are defined as the time periods surrounding and including the haze event days: Case 1 (15–18 Aug), Case 2 (20–23 Aug), and Case 3 (26–29 Aug).

On 15 Aug, prior to the onset of the first haze event in Colorado, the plume of enhanced AOD propagating from the fires in the Pacific Northwest remained north of Colorado in Montana and southern Canada (Figure 4). The air above the Denver/Boulder area contained relatively diminished AOD (0.12, averaged from the domain of 39.5°N, 104.5°W, 40.5°N, and 105.5°W). Although the core of the plume remained north of Colorado, its more diffuse southern region drifted south-eastward on 16 Aug. By 17 Aug, enhanced AOD was observed along the Front Range in northcentral Colorado near Denver/Boulder (0.37). The AOD slightly decreased on 18 Aug over Denver/Boulder (0.25), which is supported by the decrease of PM starting on 18 Aug from the CDPHE data (Figure 3). AOD increased in value and spatial extent on 20 Aug during the second haze event, when more fires were detected in the Pacific Northwest (see



increase in number of MODIS hot spots in Figure 5). This plume contained a high density of aerosols travelled over
the northcentral U.S. The southern periphery of this plume impacted Colorado east of the Continental Divide starting
on 20 Aug, as corroborated by the CDPHE air quality measurements in Figure 3. Although the AOD values were not
as enhanced over Colorado as compared to the core of the AOD plume, AOD values over the Front Range were
enhanced as compared to before the long-range transport of this plume. Enhanced AOD was observed around
Denver/Boulder and the Front Range the following three days (0.26–0.35), with the largest values in this four-day
period observed on 23 Aug. The third haze event (Figure 6) followed a similar evolution to the first two. The AOD
plume remained north of Colorado on 26–27 Aug, then infiltrated the northern and eastern part of the state on 28–29
Aug. The AOD values over Denver/Boulder during this event (0.26–0.45) were considerably larger than the two
previous events. It is important to note that AOD is a column measurement, thus the largest aerosol concentrations
may be elevated in the atmosphere as compared to what is observed on the ground. However, the AOD observations
still provide information regarding the spatial extent of the plume of aerosols emitted from the fires and that Colorado
was indeed impacted by air transported from the Pacific Northwest fires.

Further, the satellite retrievals generally corroborate the air quality observations on the ground along the Front Range
in terms of when large concentrations of aerosols might be expected. More fires were detected across the Pacific
Northwest by MODIS during the second event (678 fires, on average) when $PM_{2.5}$ was largest as compared to the first
event (231 fires, on average), which had the smallest maximum $PM_{2.5}$ out of the three haze events. The third event
had $PM_{2.5}$ values in between the first and second, while also having 607 fires on average. Thus, the number of fires
likely influenced the relative amount of smoke produced and transported to the Front Range. However, meteorological
conditions as described below also played a vital role in enabling transport of the smoke.
**3.3 Synoptic and regional scale meteorology fuel long-range aerosol transport from the Pacific Northwest**
The transport of the enhanced AOD plume from the Pacific Northwest to Colorado during each of the three events,
and the relationship between the AOD column and ground-based in situ observations, are supported by the
meteorological features present on both the synoptic and regional scales. Plan-view synoptic analyses aloft and at the
surface during the first air quality event along Colorado's Front Range on 17–18 Aug 2015 are shown in Figure 7. At
500 hPa (Figure 7a and c), a transient shortwave trough embedded in baroclinic zonal flow aloft migrates eastward
across the northern Rocky Mountains (i.e., north of Colorado), with westerly (north-westerly) flow preceding
(following) the passage of the trough axis. These flow patterns are corroborated by the HYSPLIT air mass back
trajectories during the first event, shown in Figure 8a. At the surface, high pressure and shallow cool air initially
resides primarily north of Colorado at 0600 UTC 17 Aug (Figure 7b). However, by 2100 UTC 17 Aug (Figure 7d),
the shallow cool air has moved southward across eastern Colorado. A companion time-height section of hourly wind
profiles at BAO (Figure 9) shows low-level southerly flow ahead of the frontal passage at ~1100 UTC 17 Aug and
generally west to northwest flow aloft for the duration of the plot. The observed flow aloft is represented in many of
the back trajectories, which show west to northwest flow reaching Boulder during this event. Following the frontal
passage at the wind profiler, the shallow cool air mass deepens to ~3 km MSL by 1800 UTC 17 Aug in generally





northerly-component flow. Thereafter, the depth of the cool air decreases as the low-level flow shifts to south-easterly.
Operational rawinsonde data from Denver (not shown) captures the top of the frontal inversion at 2.1 km MSL at 1200
UTC 17 Aug and at 2.7 km MSL at 0000 UTC 18 Aug, consistent with the wind-profiler analysis of the time-varying
frontal altitude at BAO. For plan-view context, the times of the synoptic analyses are marked on the time-height
section. The large $PM_{2.5}$ values (Figure 3) on 17 Aug are corroborated by the transition of air arriving from enhanced
AOD regions (see air mass backward trajectories in Figure 8a) over and off the coast of the Pacific Northwest and
northern California (Figure 4c). $PM_{2.5}$ increased markedly after the passage of the shallow front, thus suggesting the
post-frontal air mass—which originated over Wyoming downstream of the Pacific Northwest fires—contains a large
concentration of particulates from those fires.

The evolution of the shallow cold front described above is typical of southward propagating cold fronts more generally
across eastern Colorado, and the frontal propagation is influenced heavily by the complex regional topography
depicted in Figure 1. Specifically, the blocking effect of the Rocky Mountains accelerates cold air southward along
the eastern side of the high terrain (e.g., Colle and Mass, 1995; Neiman et al., 2001). Additionally, the postfrontal
northerly-component airstream flowing across the west-east-oriented Cheyenne Ridge in south eastern Wyoming
induces an anticyclonic gyre to the lee (south) of this ridge, subsequently shifting the postfrontal flow from northerly
to easterly and driving the front westward against Colorado's Front Range (e.g., Davis, 1997; Neiman et al., 2001).

The meteorology during the second air quality event, on 22-23 Aug (Figure 10), is qualitatively similar to its
predecessor, although the transient shortwave trough aloft is more amplified during the latter event (Figure 10a and
c). Consequently, during the second event, the terrain-trapped cold front and its trailing shallow cool air mass east of
the Rockies surges much farther southward across eastern New Mexico (Figure 10b and d). The corresponding air
mass back trajectories (Figure 8b) travel south-eastward from the Pacific Northwest fires to Colorado. The wind-
profiler analysis at BAO (Figure 11) shows an abrupt low-level wind shift from westerly to easterly with the frontal
passage at 1900 UTC 22 Aug, followed by a rapid deepening of the shallow cool air mass to nearly 3 km MSL.
Thereafter, the depth of this air mass ranges between ~2.2 and 3.4 km MSL. Nearby rawinsonde observations at
Denver from 0000 UTC 23 Aug to 0000 UTC 24 Aug (not shown) document a strong frontal inversion ranging
between 3.3 and 3.8 km MSL, consistent with the wind-profiler analysis. Above the shallow cool air mass, the profiler
shows westerly flow aloft, shifting to north-westerly with the passage of the transient shortwave trough. The largest
$PM_{2.5}$ values observed during this event, on 23 Aug, corresponds to the most direct transport of air (Figure 8b) from
over the enhanced AOD regions over the Pacific Northwest fires (Figure 5). As with the previous case, the $PM_{2.5}$
increased markedly with the passage of the shallow front (Figure 3). Significantly, air quality is considerably poorer
with the second event, perhaps due partly to a stronger cold-frontal push across Colorado's Front Range that originated
near the smoky source region and partly due to north-westerly (rather than westerly) flow aloft that could transport
the smoke through a deeper layer toward Colorado. Further, more fires were detected during the second event (678,
on average) compared to the first event (231 fires, on average), thus the larger number of fires could result in more
smoke production and thus a denser smoke plume transported to the Front Range.






The synoptic-scale conditions on 27–28 Aug (Figure 12) associated with the third air quality case differ significantly
from those of the two earlier events. Most significantly, a broad ridge aloft covers the intermountain West for the
duration of this final event, while an embedded weak shortwave trough migrates eastward through the ridge from
Wyoming/Colorado to the Great Plains (Figure 12a and c). A surface reflection of the upper-level shortwave trough
is manifest as a weak low-pressure centre over western Nebraska and Kansas at 1800 UTC 27 Aug (Figure 12b). This
low migrates eastward during the subsequent 24 h (Figure 12d) in tandem with the upper-level shortwave. Because
this surface low resides beneath a mean ridge aloft, the temperature contrast across this trailing cold front is weaker
than its earlier counterparts (not shown). Nevertheless, the southward migration of the front east of the Rockies
suggests that terrain blocking may have influenced its evolution. The air mass back trajectories show parcels
originating from the region of the fires, similar to the trajectories from the earlier two events. Companion observations
from the BAO wind profiler (Figure 13) capture the shallow frontal passage at 2000 UTC 27 Aug, when westerly flow
shifts abruptly to northerly. Above 3 km MSL, the wind field exhibits a more gradual transition from westerly to
north-westerly as the weak shortwave trough moves across the wind profiler. The Denver rawinsondes at 0000 and
1200 UTC 28 Aug observed a frontal inversion at ~2.1 km MSL (not shown). It is less prominent than the frontal
inversions during the earlier events, largely because the temperature contrast across this front is weaker than its
predecessors. The subsequent rawinsonde profile at 0000 UTC 29 Aug (not shown) captures a deep, dry-convective
boundary layer extending up to 4 km MSL, despite persistent low-level northerly flow. Sensible heating eroded the
remnant low-level cool air east of the Rockies. $PM_{2.5}$ increases following the initial shallow cold-frontal passage at
2000 UTC 27 Aug and continues to increase for the remainder of the wind-profiler time-height section, as deep
northerly-component flow behind the weak shortwave trough transports smoke particulates across Colorado.
**3.4 Mineral dust and smoke arrive along the Front Range**
The types of aerosols present in the enhanced AOD plumes that were transported towards the Front Range via the
aforementioned synoptic conditions were evaluated using additional satellite-based measurements and support the
interpretation of transport of aerosols from the wildfires in the Pacific Northwest to Colorado. Figures 14–16 show
aerosol subtype data from the CALIPSO satellite in planar (a panels) and vertical-profile (b panels) views. Only on
the day prior to, or on the worst day of, each haze event are shown, although aerosol subtype data were examined
anytime CALIPSO passed over the Pacific Northwest or Colorado from 15 Aug to 2 Sep. CALIPSO demonstrates the
presence of smoke, dust, or polluted dust (dust mixed with smoke in each profile) during times that intersect the
enhanced AOD plume propagating from the Pacific Northwest or when over Colorado. Dust and smoke plumes from
the fires extended up to 10 km MSL over the western U.S. The mineral dust and smoke detected by CALIPSO in
transit to the Front Range was also detected with the TOPAZ lidar and the in situ aerosol particle mass and speciation
monitor at the DSRC. Figure 17 shows aerosol extinction profiles from the surface to 2.5 km AGL measured with the
TOPAZ lidar on 9 days during the smoke episodes. The time resolution of the extinction profiles is 5 minutes and the
vertical resolution is 1 m at the lowest altitudes, increasing to 6 m above 500 m AGL. The observations on 14 Aug
and 2 Sep, which bracket the smoke episodes, indicate very clean conditions with AOD from the surface up to 2.5 km



AGL ($AOD_{2.5km}$) of 0.05 and 0.04, respectively. Aerosol extinction coefficients and $AOD_{2.5km}$ were significantly
larger during the smoke episodes with an approximately 7-fold increase in $AOD_{2.5km}$ on 20 and 21 Aug. Aerosol
extinction was enhanced over the entire 2.5 km column, but the largest aerosol extinction values were observed in the
boundary layer in the lowest few hundred meters up to 1.5 km AGL. Also, the lidar measurements reveal that on most
days aerosol extinction varied significantly over the course of the day (e.g. 20 Aug). The largest aerosol extinction
values around 1–1.5 km AGL observed on 19 Aug were primarily due to swelling of aerosol particles in the moist
relative humidity environment beneath cumulus clouds at the top of the boundary layer. However, aerosol extinction
in the lower part of the boundary was still significantly larger than on 14 Aug, which is consistent with the larger
aerosol particle concentrations in the smoke plumes. The lidar measurements are consistent with the in situ $PM_{2.5}$ and
MODIS AOD observations. When comparing lidar $AOD_{2.5km}$ with MODIS AOD one has to be cognizant of the fact
that the TOPAZ observations only cover a portion of the atmospheric column and that the two AOD measurements
were made at different wavelengths.

Figure 18 shows the time series of $PM_{2.5}$, soil mass concentrations, and elemental mass concentrations (data from the
PX-375 was not available prior to this time period due to instrumental complications). Soil concentrations were
calculated by following the Interagency Monitoring of Protected Visual Environments (IMPROVE) convention using
concentrations of specific metals: SOIL = 2.2[Al] + 2.49[Si] + 1.63[Ca] + 2.42[Fe] + 1.94[Ti] (Malm et al., 1994;
Hand et al., 2011). Both $PM_{2.5}$ and soil mass concentrations increased during the worst haze event days (i.e., 26 and
29 Aug), when the Pacific Northwest fires were influencing air along the Front Range and when CALIPSO showed
the presence of smoke and dust over the western U.S. The diurnal pattern is likely caused by the upslope/downslope
flow patterns due to proximity from the base of the foothills, which is particularly pronounced in the summer (Toth
and Johnson, 1985). Further, select metals also increased in concentrations during haze events, particularly those
typically sourced from mineral dust (i.e., in the IMPROVE soil convention equation) and S and K, which are metal
tracers that have been observed in smoke or biomass burning aerosols originating from fires (Artaxo et al., 1994;
Gaudichet et al., 1995; Yamasoe et al., 2000; Pachon et al., 2013).

Figure 19 shows the average concentrations of mineral dust or biomass burning metal tracers from the PX-375 from
26 Aug to 2 Sep, during conditions influenced by the Pacific Northwest fires (days with enhanced $PM_{2.5}$; 29–30 Aug)
and days with cleaner, normal Front Range conditions (days with small $PM_{2.5}$; remaining days during this time period).
$PM_{2.5}$ and soil mass, biomass burning metals (S and K), and mineral dust marker (Al, Si, Fe, and Ca) concentrations
were all larger, on average, during influences from the Pacific Northwest fires, corroborating the CALIPSO
observations. Also included are metals that are typical of industrial tracers As and Pb (Figure 19e) (Paciga and Jervis,
1976; Hutton and Symon, 1986; Thomaidis et al., 2003), which were actually smaller during influences from wildfires
and larger during normal, regionally-sourced influences. The average $PM_{2.5}$ mass concentration from the CDPHE data
was almost 3 times larger on 29–30 Aug as compared to the remaining days in the 26 Aug–2 Sep time period (15.9
versus 5.7 µg m$^{-3}$, respectively). This result demonstrates how influences from typical, regional industrial sources is
disrupted by the synoptic conditions that introduced the long-range transported biomass burning plumes. Although Zn



and Cu have been shown to originate from wildfires (Yamasoe et al., 2000), the averages were similar—within 1 ng
m$^{-3}$—thus a distinct comparison could not be made within certainty. Further, these metals can also be derived from
vehicular emissions, thus their concentrations may additionally be influenced by local traffic (Sternbeck et al., 2002).
These results demonstrate the transport of mineral dust and biomass burning aerosol species to the Front Range, which
were indeed larger in concentration during poor air quality/haze events. Interestingly, mineral dust mixed within a
smoke plume from fires has predominantly been observed originating from more arid regions along the global dust
belt, and using modelling or remote sensing data only (e.g., Radojevic, 2003; Tesche et al., 2009; Yang et al., 2013;
Nisantzi et al., 2014). To our knowledge, this co-lofting of dust and smoke has not been shown to occur in the U.S.,
particularly in a region as densely covered in vegetation as the Pacific Northwest.

**4 Conclusions**

We have demonstrated the transport of large quantities of mineral dust and smoke/biomass burning aerosols from
wildfires in the Pacific Northwest to the Colorado Front Range. These aerosols were transported under synoptic
conditions that contributed to three different haze events, inducing poor air quality in the Denver metro area. Three
separate poor air quality events with enhanced PM$_{2.5}$ were likely dependent on the number of fires and observed to
occur with cold frontal passages along Colorado's Front Range, enabling the enhanced AOD plumes originating from
the Pacific Northwest wildfires to propagate south-eastward to Colorado's Front Range. Air masses were shown to
originate from over the region dense with wildfires, and followed through satellite-detected aerosol plumes, which
were rich in a mixture of mineral dust and smoke. Tracers for these aerosol types were also detected in situ along the
Front Range, and were shown to be enhanced during periods of influence from the fires.
Overall, these unique observations were demonstrated using a complete suite of in situ and remote sensing aerosol
measurements in the context of in situ meteorological observations and air mass trajectory modelling. In tandem, we
utilized a real-time X-ray fluorescence spectroscopy technique using the novel and field-portable PX-375 from
HORIBA, Ltd. demonstrating the utility of the instrument. Although the haze events were short lived, they
demonstrate how quickly aerosols can be transported long distances and affect air quality in regions thousands of
kilometres away. Interestingly, mineral dust was observed to be co-lofted and transported within the smoke plumes,
an observation not previously reported for vegetated regions such as the Pacific Northwest.
Mineral dust and smoke aerosols have disparate implications for health and climate (i.e., by serving as seeds for cloud
particle formation, which impacts cloud lifetime, radiative effects, and precipitation formation mechanisms),
particularly at the levels observed along the Front Range. These unique observations should be taken into account
when developing health standards, seeing as not only regional urban and industrial emissions contribute to poor air
quality conditions. Additionally, dust and smoke are efficient cloud forming nuclei, particularly when orographically
lifted along barriers such as the Front Range into the upper atmosphere, where cloud formation is prominent. Thus,
transport of these aerosols from wildfires has broad implications for altering aerosol composition in regions far from
the source.





**Author contribution.** J. M. C. analysed XRF data, compiled CDPHE and MODIS data, ran HYSPLIT simulations,
and wrote the manuscript. P. J. N. conducted meteorological analysis and interpretation. T. C. compiled and analysed
CALIPSO data. C. J. S., G. K., and R. A. analysed and supplied TOPAZ data. A. Y. provided PX-375 for usage. All
co-authors contributed to the writing of or provided comments for manuscript.
**Acknowledgements.** The authors would like to acknowledge the many agencies and organization from which data
were acquired, including the CDPHE for air quality data, NASA for MODIS and CALIPSO observations, NOAA for
HYSPLIT and HMT meteorological data, and the HDF group for providing example code to process CALIPSO data.

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

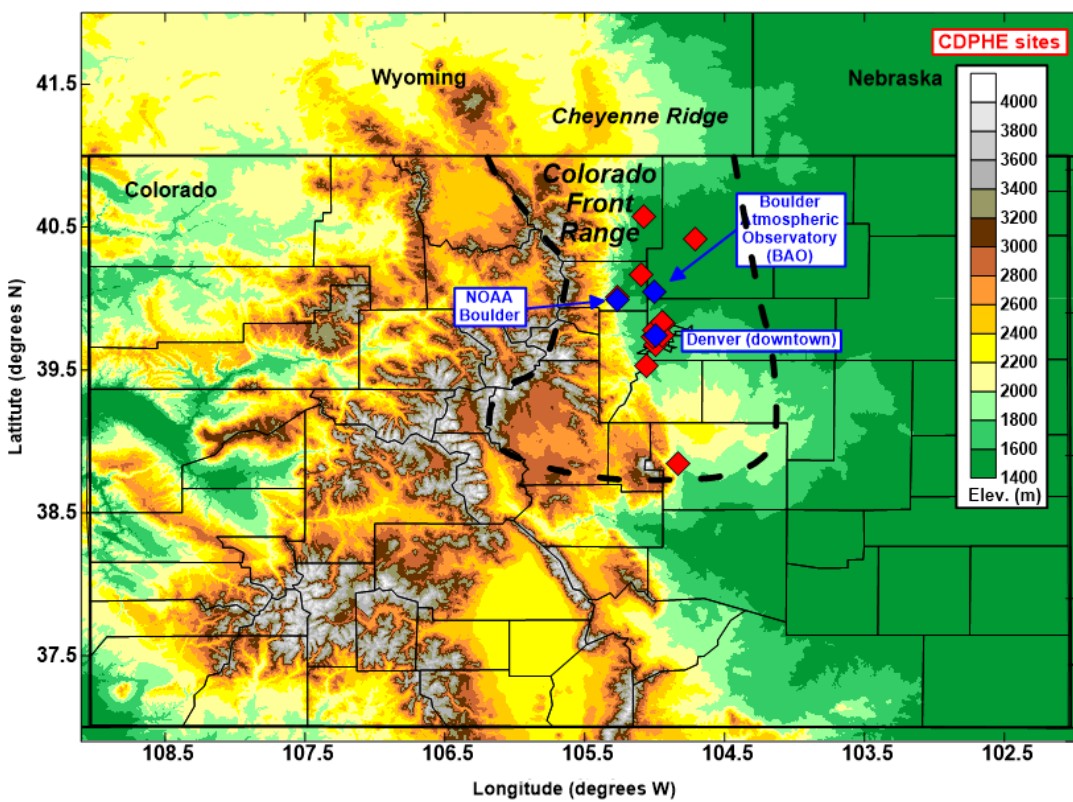


**Figure 1.** Map of monitoring locations, including NOAA DSRC in Boulder, which housed the PX-375 and
TOPAZ lidar instruments, the BAO where the 449-MHz wind profiler was deployed, downtown Denver, and
the CDPHE sites where $PM_{2.5}$ and $PM_{10}$ are monitored (see Table 1 for site descriptions). The approximate
area encompassing the Colorado Front Range is highlighted by the dashed line. The Cheyenne Ridge in
Wyoming is also notated.



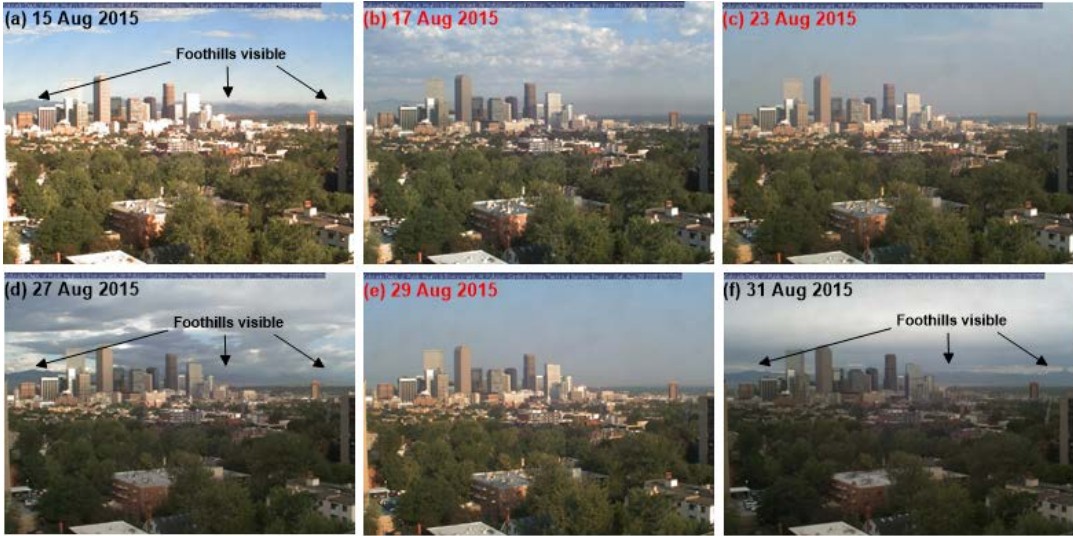

553

**Figure 2. Images of downtown Denver facing west taken at 1400 UTC (0800 MDT). Images acquired from the CDPHE**

**Visibility Station (DESCI; 39.73°N, 104.96°W; 1633 m MSL). Only days of significant meteorological and visibility**

**transitions in August 2015 are shown. Days in red are those which correspond to the haziest days during the study time**

**period. In panels (a), (d), and (f), the visibility of the foothills (and background high terrain) is highlighted.**





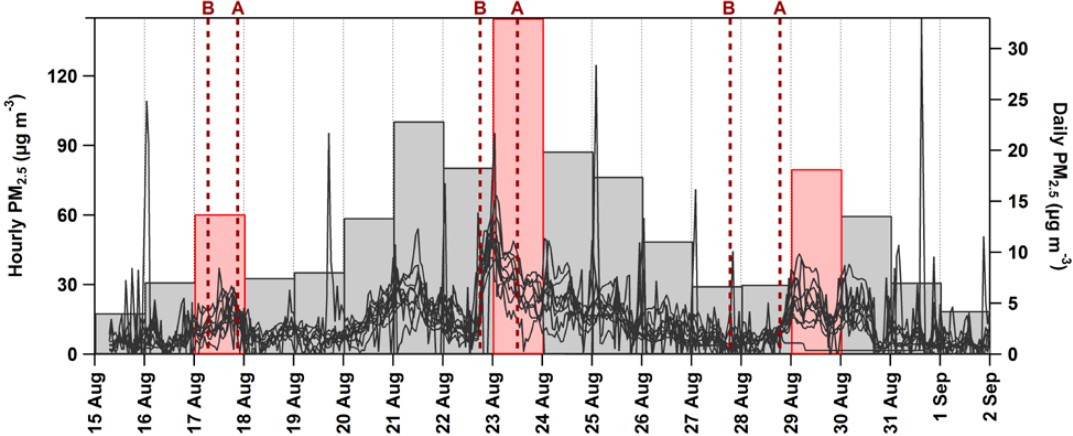


**Figure 3. Hourly and daily averaged PM$_{2.5}$ mass concentrations at CDPHE sites. The pairs of red dashed lines**
**shows the times before "B" and after "A" cold-frontal passages at BAO during or prior to each haze event. The**
**daily averaged PM$_{2.5}$ in red represent the haziest days during or following cold front al passages (i.e., Events**
**1, 2, and 3 on 17, 23, and 29 Aug 2015, respectively).**




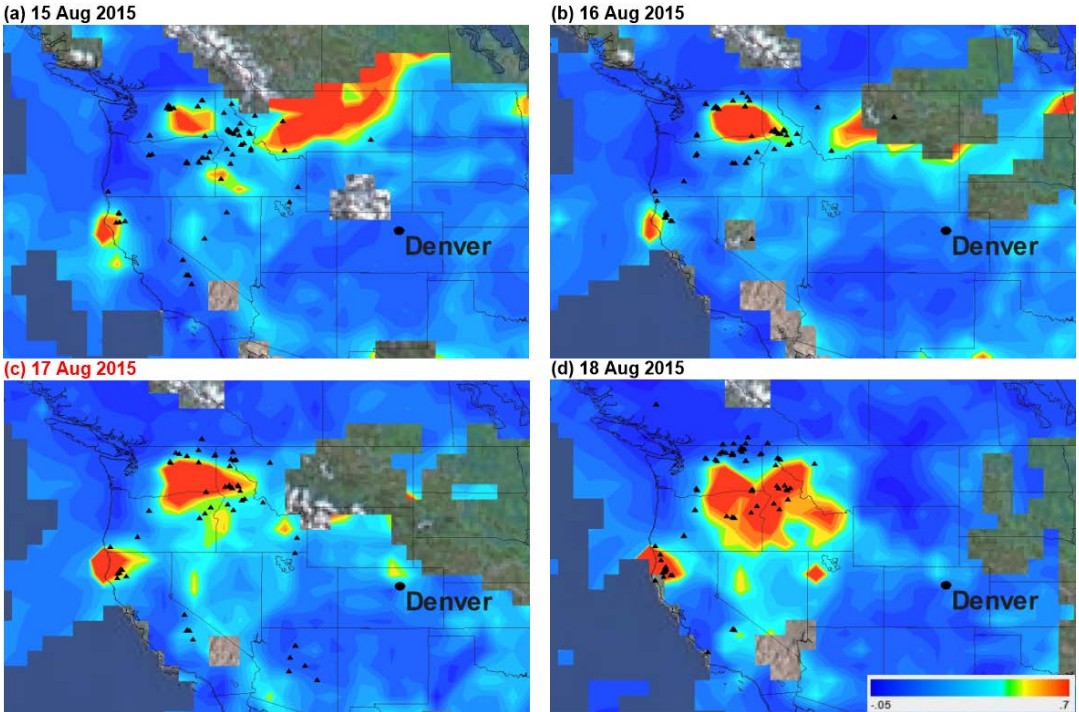


**Figure 4. Daily averaged aerosol optical depth (AOD; colour bar lower right) and fire hotspots (black markers)**

**detected by MODIS during the first major haze case study between 15 and 18 Aug 2015. The haziest day from**

**the CDPHE data is labelled in red (i.e., Event 1).**





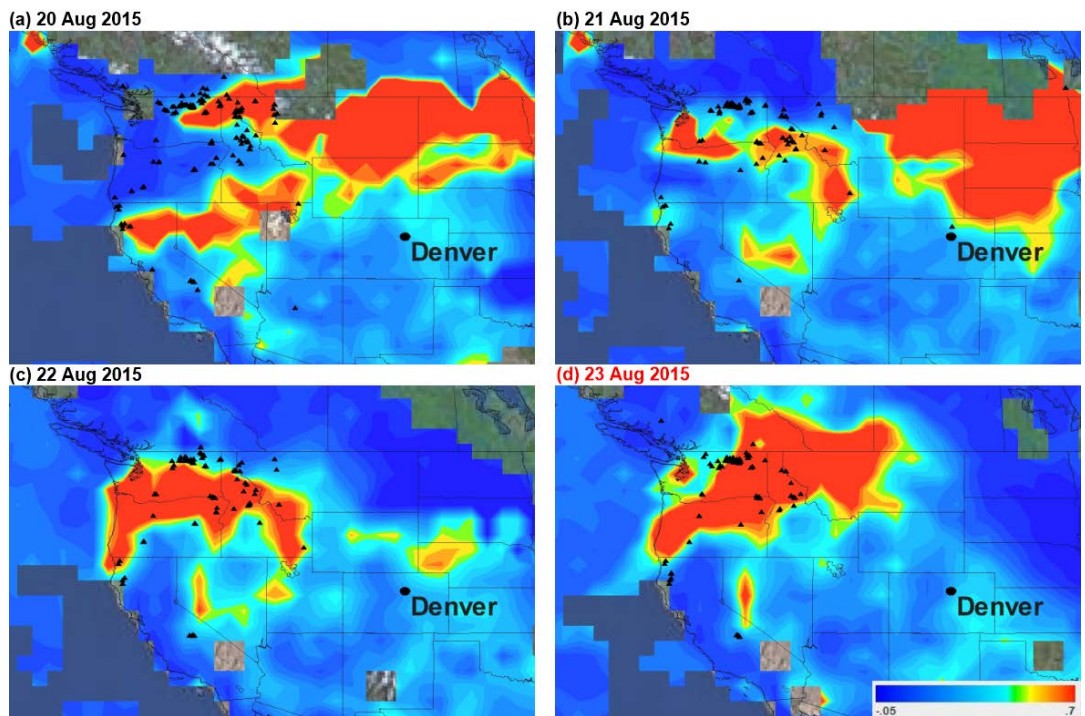


**Figure 5. Same as Figure 4, but for the second major haze event between 20 and 23 Aug 2015. The haziest day**

**from the CDPHE data is labelled in red (i.e., Event 2).**



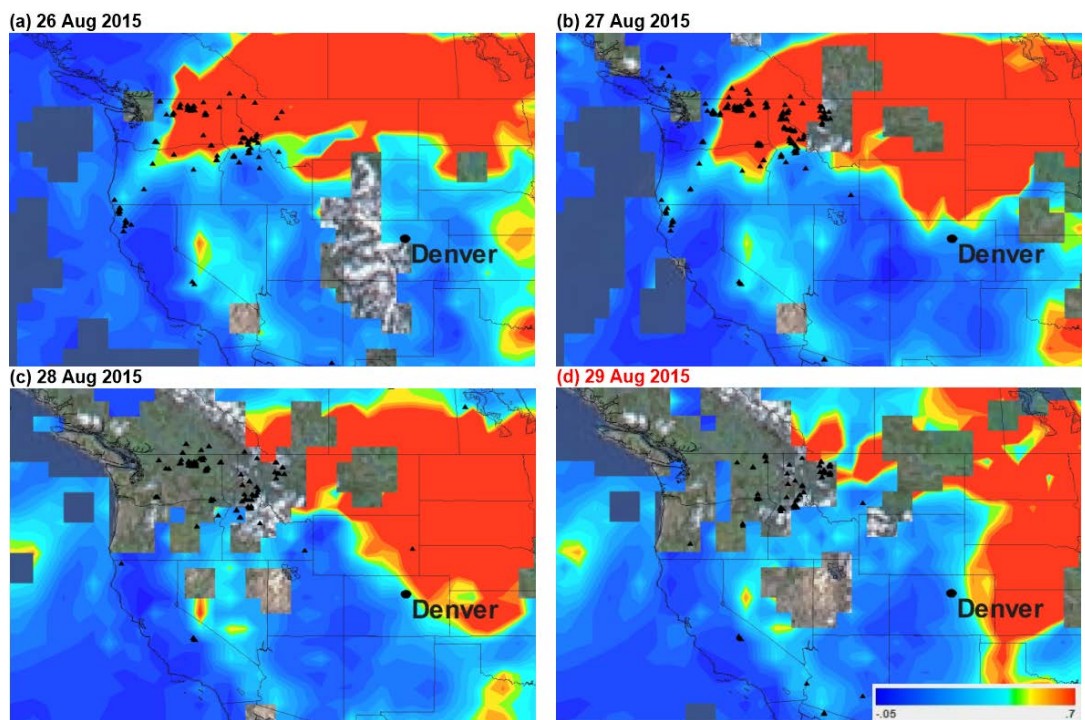


**Figure 6. Same as Figure 4, but for the third major haze event between 26 and 29 Aug 2015. The haziest day**

**from the CDPHE data is labelled in red (i.e., Event 3).**



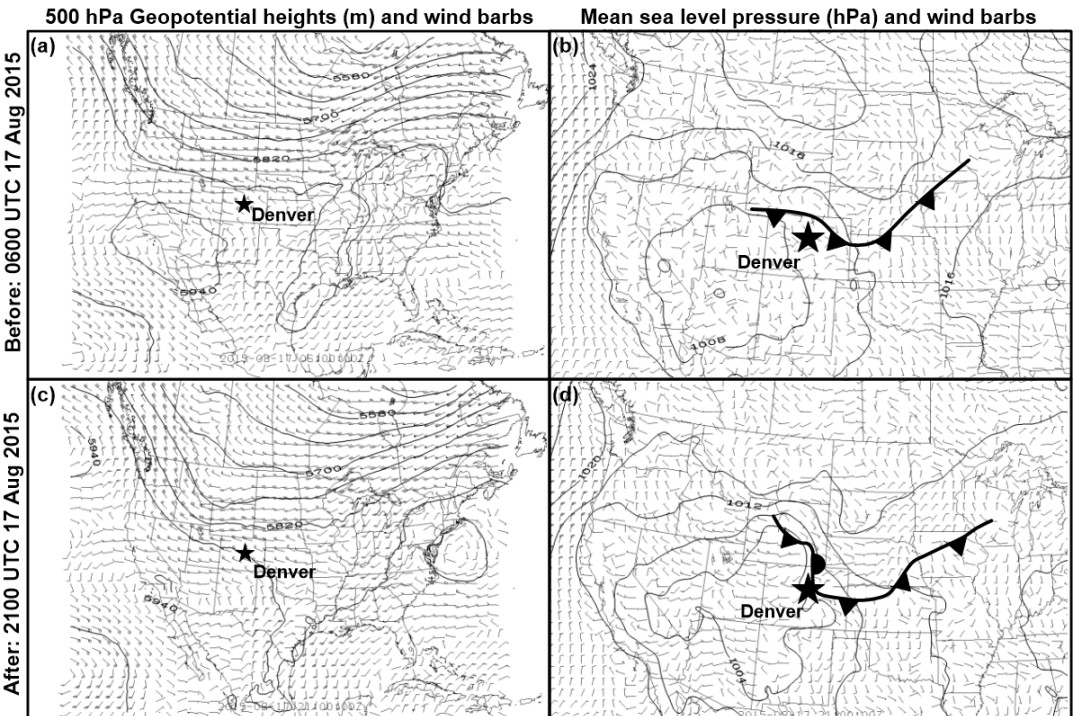

**Figure 7. Plan-view analyses from before and after the passage of a cold front during Event 1: 0600 and 2100 UTC 17 Aug**
**2015, respectively. Analyses include the 13-km resolution RAP gridded dataset of (left column) 500-hPa geopotential heights**
**(m, black contours) with 500-hPa wind velocities (flags = 25 m s$^{-1}$, barbs = 5 m s$^{-1}$, half-barbs = 2.5 m s$^{-1}$) and (right column)**
**mean sea-level pressure (mb, black contours) with near-surface wind velocities (flags and barbs as above). Standard frontal**
**notation is used.**

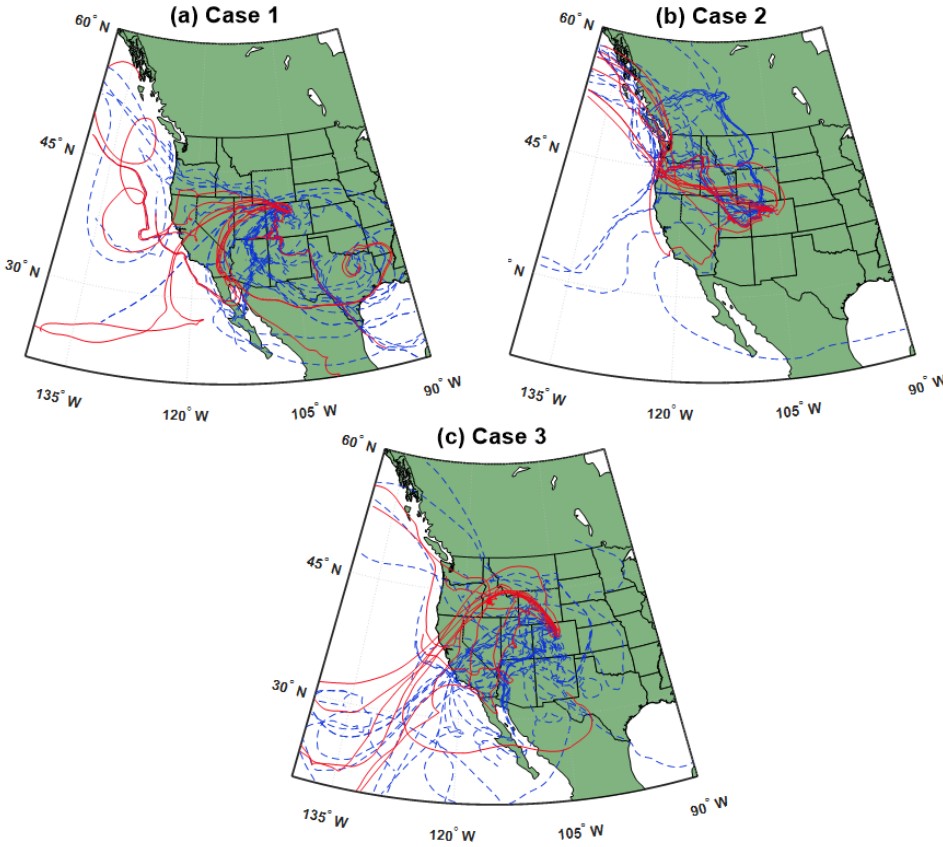


**Figure 8. Air mass backward trajectories for all three cases where haze infiltrated the Front Range. Trajectories shown**
**were initiated every 6 hours during time periods (i.e., cases) surrounding each event (15–18 Aug 2015, 20–23 Aug 2015, and**
**26–29 Aug 2015 for Cases 1, 2, and 3, respectively), include those starting at 500, 1000, and 2000 m MSL, and extend back**
**10 days. Trajectories in red correspond to the haziest days (i.e., Events 1, 2, and 3 on 17, 23, and 29 Aug 2015, respectively)**
**during each case time period and the blue dashed trajectories show the remaining days that correspond to those in the**
**MODIS figures for each case.**



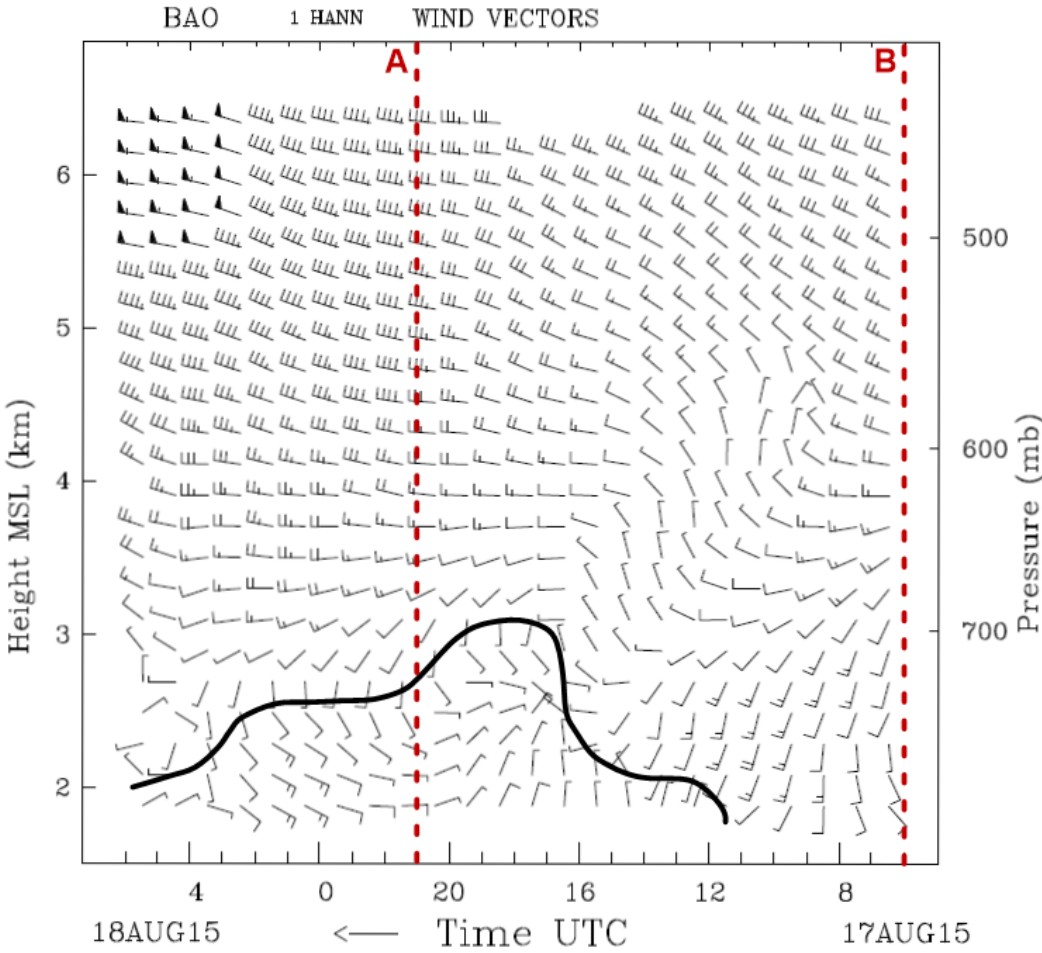

586

**Figure 9. Time-height section of hourly-averaged wind profiles for Event 1 from the 449-MHz wind profiler at BAO between**
**0600 UTC 17 Aug and 0600 UTC 18 Aug 2015 (flags and barbs are as in Figure 7). The bold black line denotes the**
**approximate frontal shear boundary. The pair of red dashed lines shows the RAP analysis times before "B" and after "A"**
**the cold-frontal passage at BAO during Event 1. Time increases from right to left to portray the advection of upper-level**
**synoptic features from west to east.**



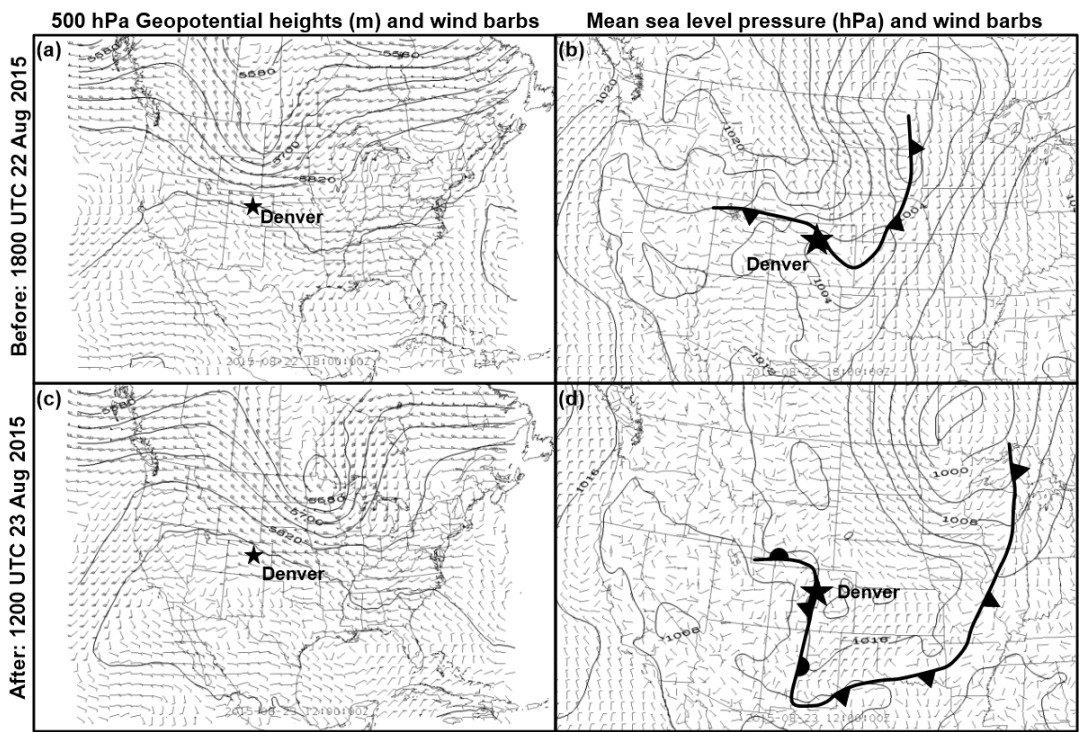

**Figure 10. Same as Figure 7, but for before and after the cold-frontal passage of Event 2: 1800 UTC 22 Aug and 1200 UTC**
**23 Aug 2015, respectively.**



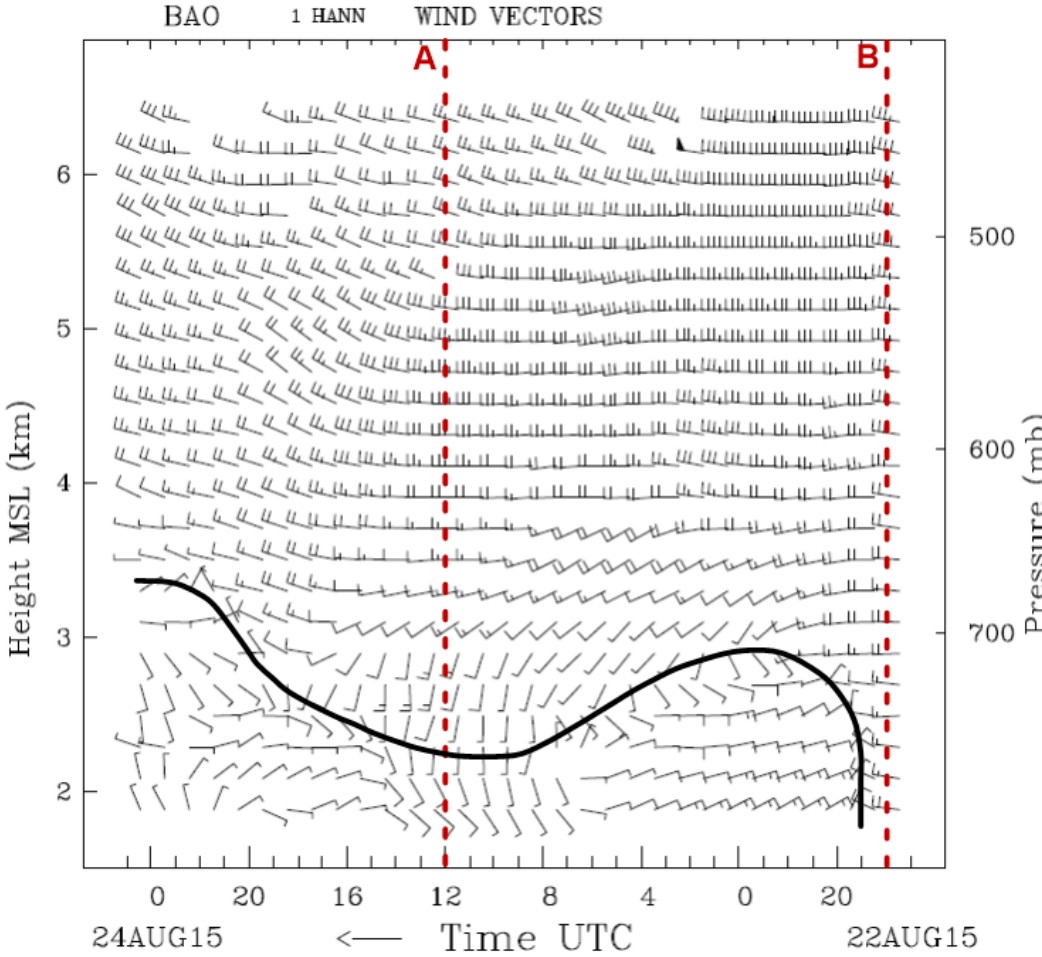

**Figure 11. As in Figure 9, but for the time period between 1700 UTC 22 Aug and 0100 UTC 24 Aug during Event 2.**



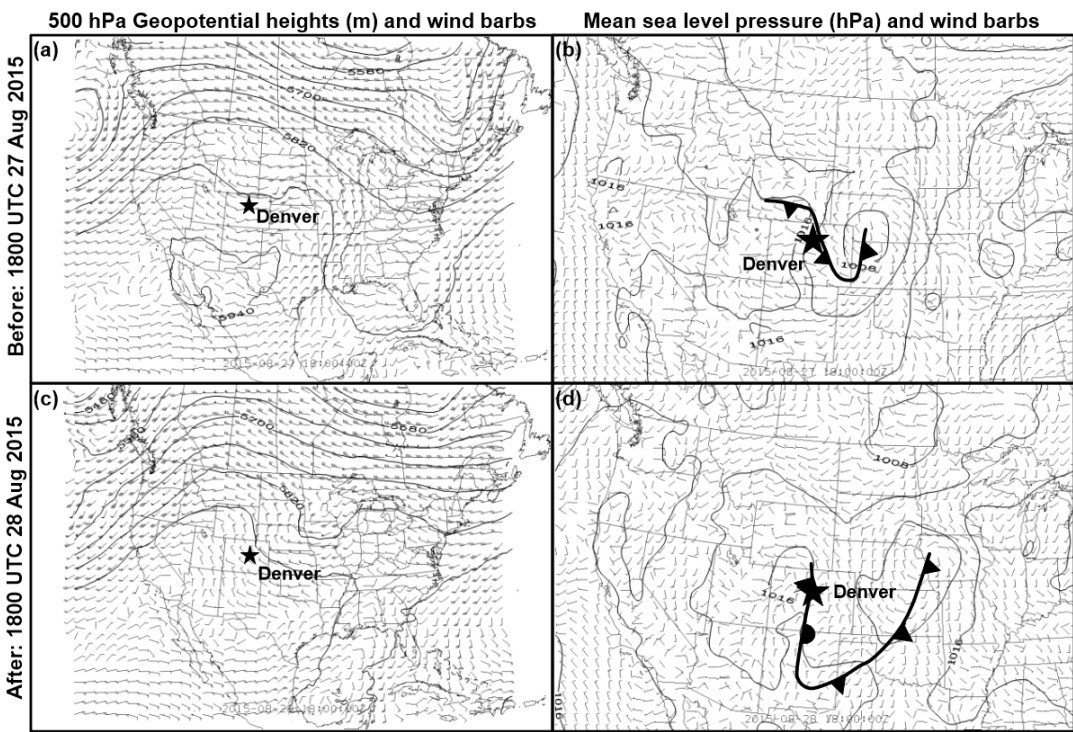

**Figure 12. As in Figure 7, but for before and after the cold-frontal passage of Event 3: 1800 UTC 27 Aug and 1800 UTC 28 Aug 2015, respectively.**



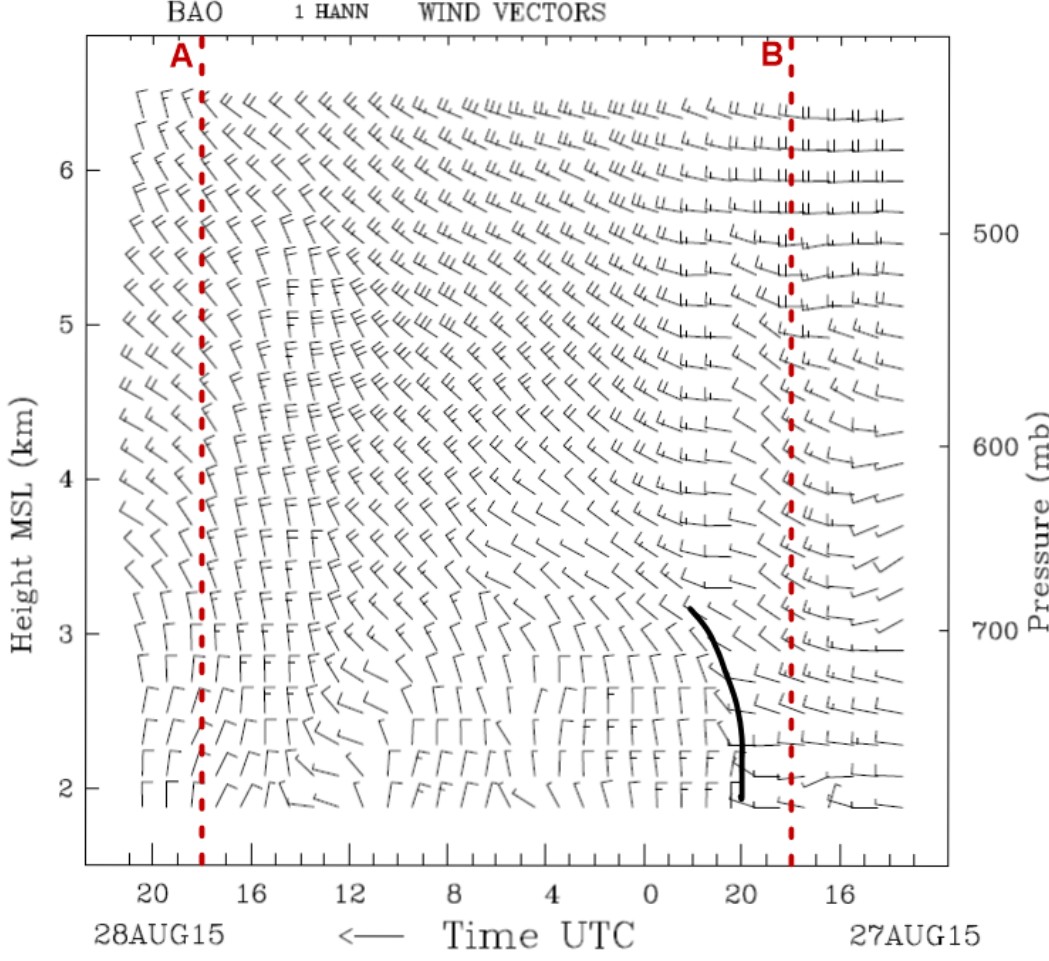


**Figure 13. As in Figure 9, but for the time period between 1300 UTC 27 Aug and 2100 UTC 28 Aug during Event 3.**




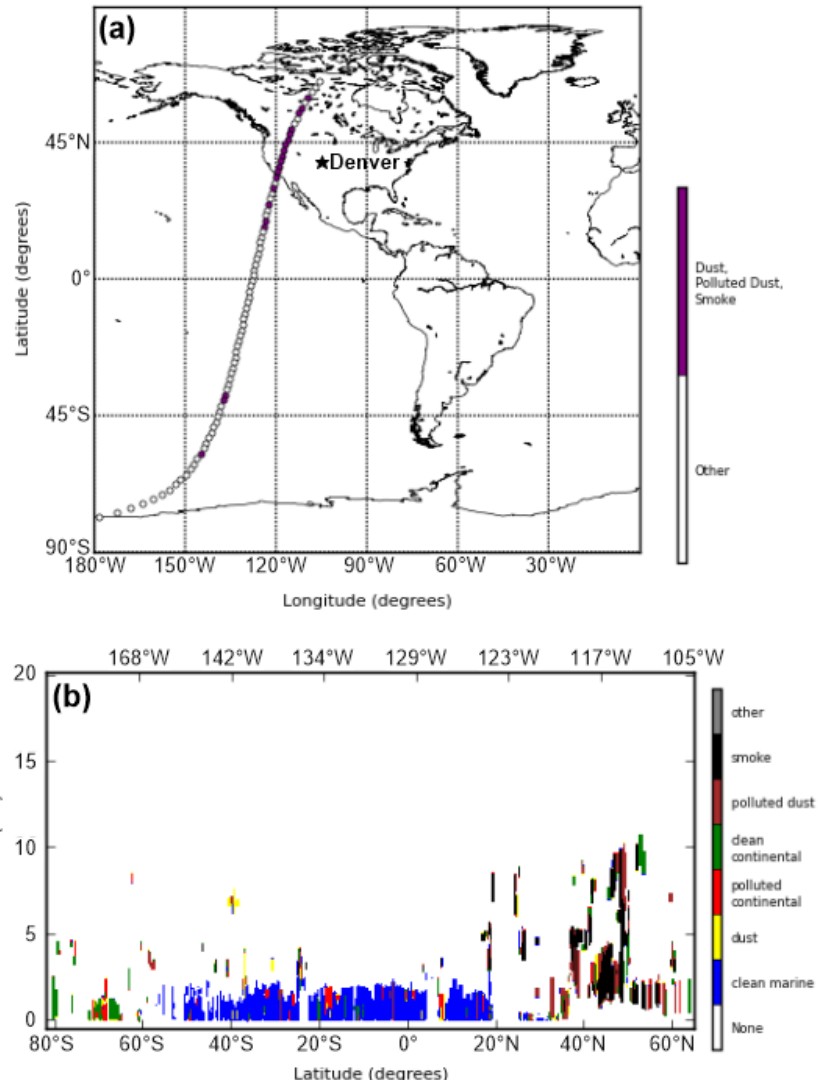

**602**

**603** **Figure 14. CALIPSO swath data from the night prior to Event 1. Swath data contained in CAL_LID_L2_VFM_ValState1-**

**604** **V3-30 file from 16 Aug 2015 09:57:00 UTC. (a) Map showing CALIPSO coverage, with the purple markers representing**

**605** **locations in the column measurement where dust, smoke, or polluted dust were observed. (b) Vertical profile (in km MSL)**

**606** **for all aerosol subtypes of the swath corresponding to (a).**




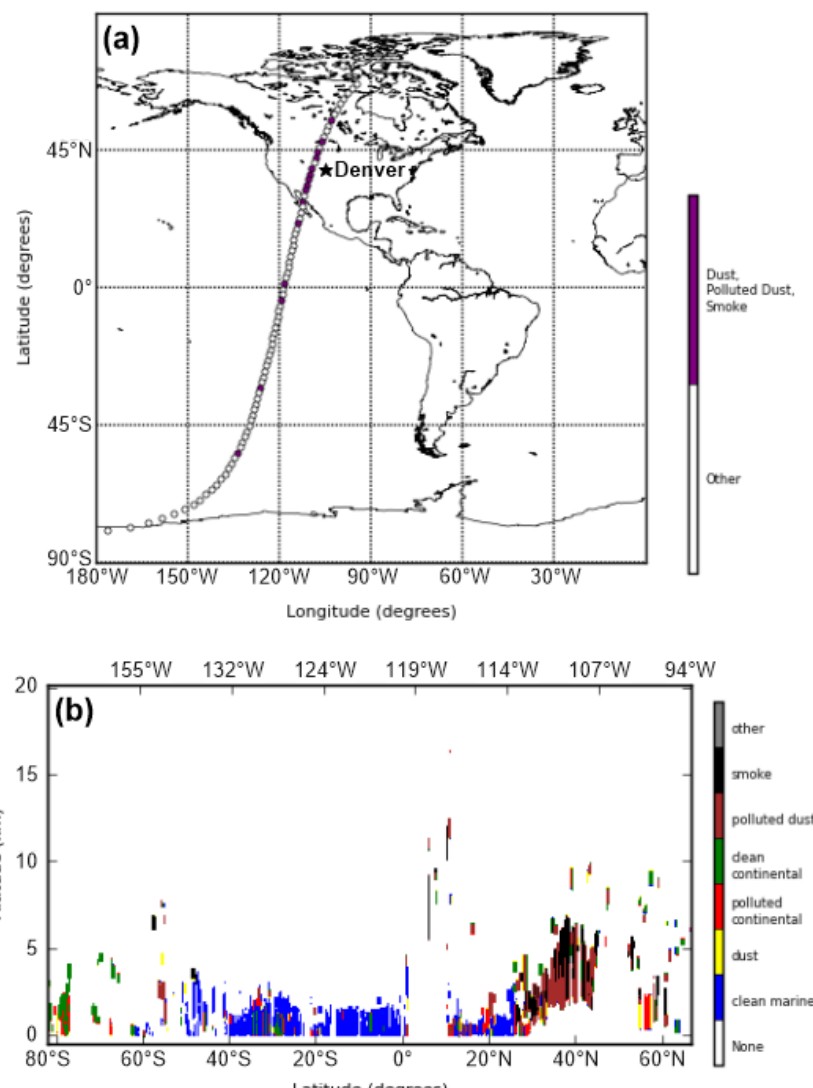

**Figure 15. Same as Figure 14, but for the night prior to Event 2 and from 22 Aug 2015 09:19:24 UTC.**



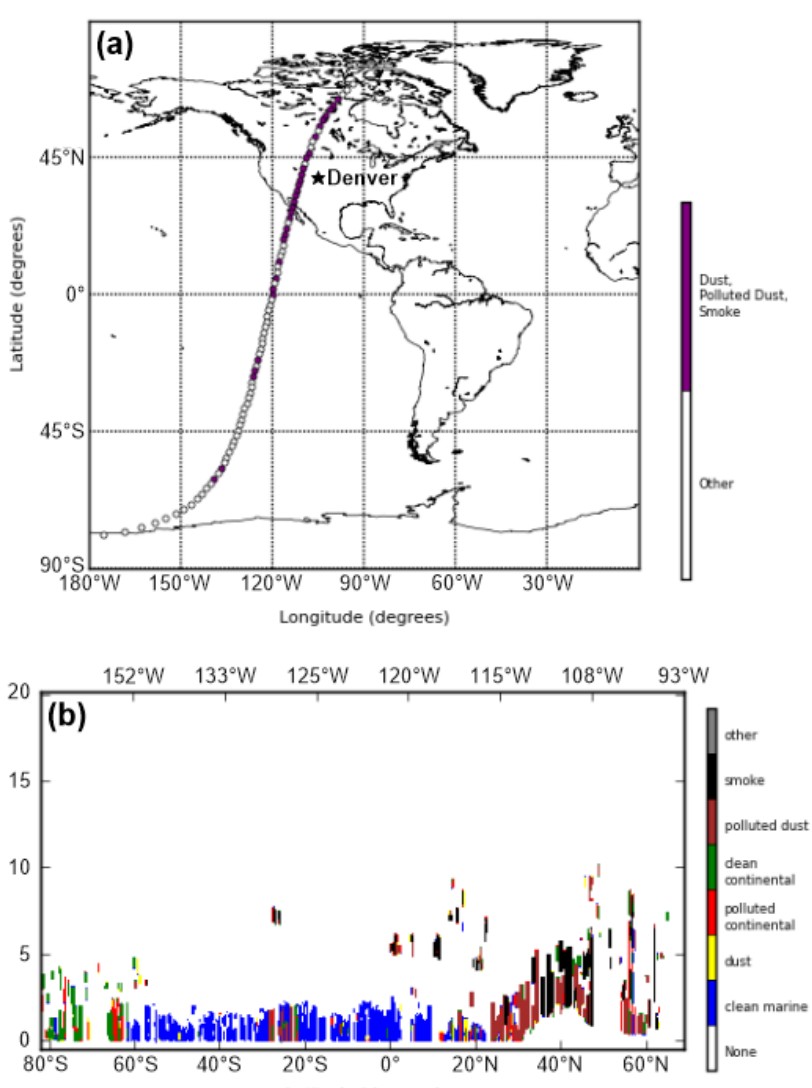


**Figure 16. Same as Figure 14, but for the day of Event 3 and from 29 Aug 2015 09:24:15 UTC.**





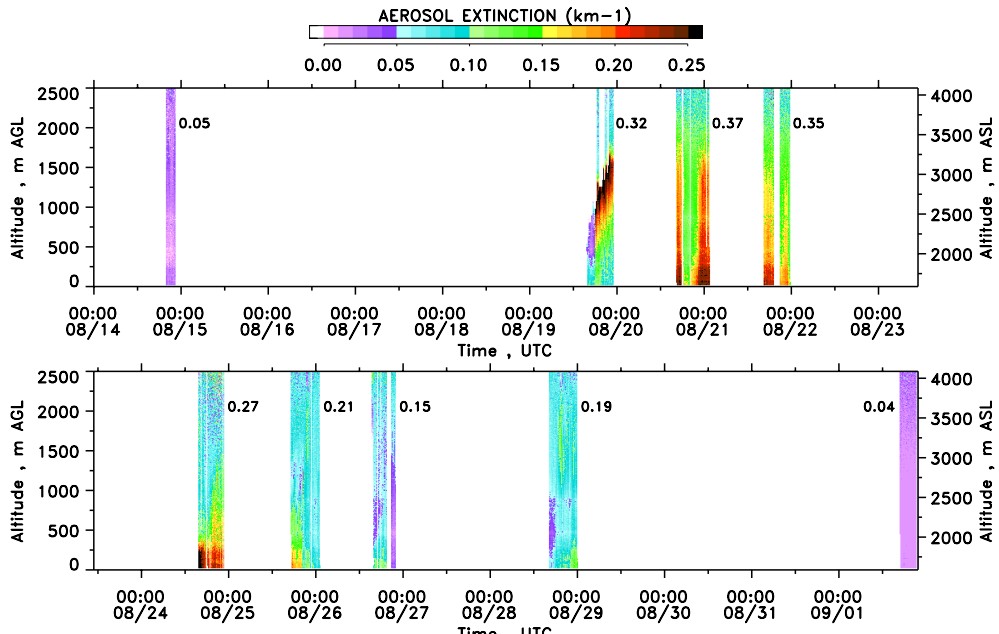


**Figure 17. Aerosol extinction profiles at 294 nm observed with the TOPAZ lidar on 9 days during the smoke pollution**

**episodes. The numbers next to each day's observations represent the daily mean AOD from the surface up to 2.5 km AGL**

**computed from the lidar measurements.**



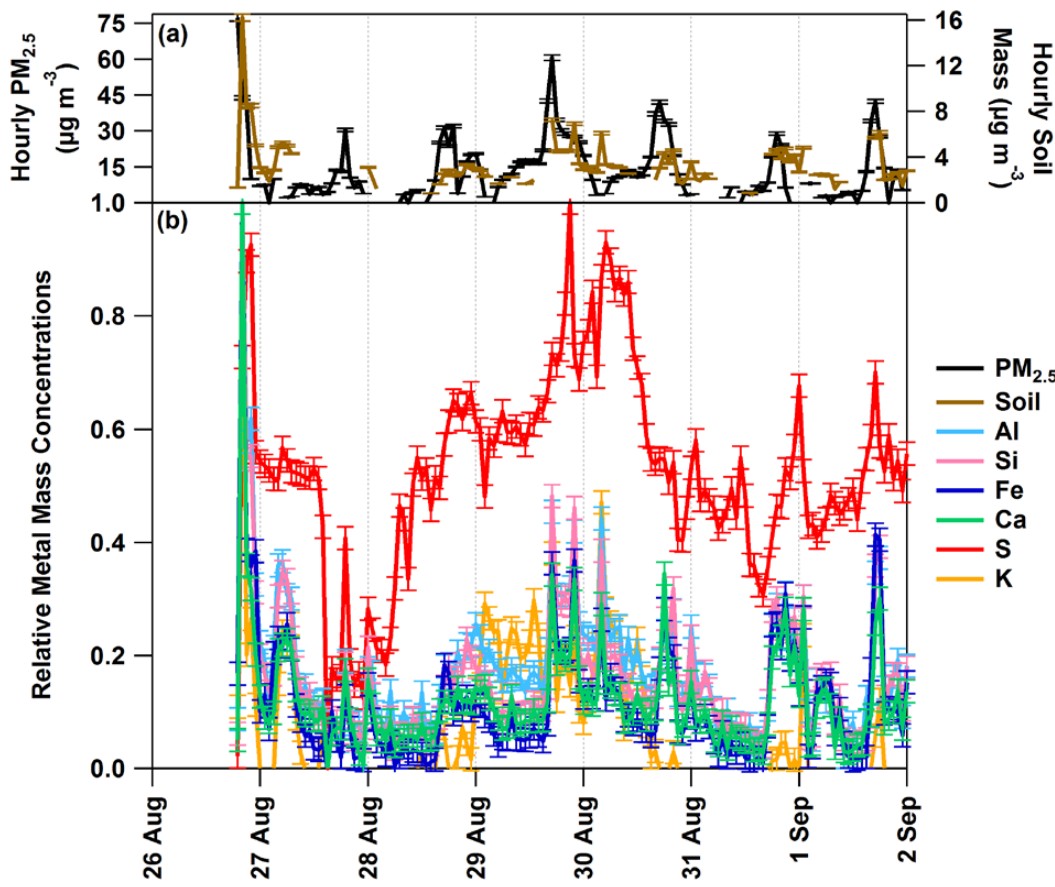


**Figure 18. (a) Time series of hourly PM$_{2.5}$ and soil mass concentrations as measured by PX-375 between 26 Aug and 2 Sep**
**2015 and (b) relative hourly mass concentrations of select individual metals over the LDLs, including an error of ±2%. PX-**
**375 data overlapped with Event 3.**




**Figure 19. Averages of PM₂.₅, soil, and select metal mass concentrations during non-event days (i.e., cleaner conditions)**
**compared to averages from haze event days (i.e., influence from fires haze) for 26 Aug–2 Sep 2015. Error bars represent**
**the 90% confidence intervals.**



**Table 1. CDPHE sites used for particulate data within the Colorado Front Range. Each site has an 'x' for each measurement it maintained throughout the current work. Elevation is provided in meters above mean sea level (m MSL).**

| City/Site Name | Site ID | Latitude (degrees N) | Longitude (degrees W) | Elevation (m MSL) | PM$_{2.5}$ | PM$_{10}$ |
|---|---|---|---|---|---|---|
| Boulder - CU/Athens | BOU | 40.01 | 105.27 | 1,621 | x | |
| Chatfield Park | CHAT | 39.53 | 105.05 | 1,685 | x | |
| Colorado College | CCOL | 38.85 | 104.83 | 1,833 | x | |
| Commerce City/Alsup Elementary | COMM | 39.83 | 104.94 | 1,565 | x | |
| Denver - Continuous Air Monitoring site | CAMP | 39.68 | 104.99 | 1,610 | x | x |
| Denver - National Jewish Health | NJH | 39.74 | 104.94 | 1,615 | x | |
| Fort Collins - CSU Facilities | FTCF | 40.57 | 105.08 | 1,525 | x | x |
| Greeley - Hospital | GREH | 40.42 | 104.71 | 1,439 | x | |
| I-25 - Denver | I-25 | 39.73 | 105.02 | 1,586 | x | x |
| La Casa | CASA | 39.78 | 105.01 | 1,601 | x | x |
| Longmont - Municipal | LNGM | 40.16 | 105.10 | 1,517 | x | |
| Welby | WBY | 39.84 | 104.95 | 1,554 | | x |

PM$_{2.5}$ = particulate matter with diameters ≤ 2.5 µm
PM$_{10}$ = particulate matter with diameters ≤ 10 µm

**Table 2. Lower detection limits (LDLs, ng m$^{-3}$) for metals measured by the PX-375 during 15 Aug–2 Sep 2015. Data less than the LDLs were excluded from analysis.**

| Species | LDL |
|---|---|
| Ti | 2.29 |
| V | 0.23 |
| Cr | 0.61 |
| Mn | 0.93 |
| Fe | 1.51 |
| Ni | 0.33 |
| Cu | 0.78 |
| Zn | 1.21 |
| As | 0.02 |
| Pb | 0.80 |
| Al | 32.2 |
| Si | 5.17 |
| S | 1.11 |
| K | 4.37 |
| Ca | 1.18 |