# Peer review of "aerosol: A set of case studies during the 2015 Pacific Northwest"

_Atmospheric Chemistry and Physics, 2016_

## Referee Comment (RC1) · Anonymous Referee #1 · 23 May 2016

This is a well-written and interesting paper documenting the transport of wildfire smoke from the Pacific Northwest into Colorado. The paper is organized around specific events that resulted in degraded air quality and visibility through the Front Range of Colorado. Supporting evidence was provided through a variety of measurement platforms, including remote sensing and ground based measurements, as well as meteorological data and back trajectory analyses to describe the flow patterns during the events. The authors have combined these data into an interesting story that informs as to the transport of smoke across the United States with impacts on local air quality. An important result is the transport of mineral aerosols with the smoke plume. I recommend the paper be published after addressing the comments below.

[Figure]

Comments Line 133: "g $\mu$m-3" typo.

Line 185: The authors repeatedly refer to "hazy" conditions along the Front Range (specifically Denver) and support the degraded air quality using PM data measured by the Colorado Department of Public Health and Environment. In checking the available data it appears that extinction data are also available from transmissometer measurements at the DESCI site. It would improve the paper to include these data so that the "hazy" can be quantified (line 185). In fact, the extinction values agree fairly closely with the TOPAZ lidar data in Figure 17 (given the wavelength differences).

Line 205: Please provide wavelength.

Line 232, Section 3.3: This section is somewhat hard to follow because the figures are broken up so it requires flipping back and forth. I suggest organizing the figures so that, for example: the first event would include figure 7a-d, 8a, 9. It would reduce the number of figures and help to focus the discussion.

Line 244,245: Do the authors mean "northwesterly" here?

Line 308: Figures 14-16 are similar enough it might be possible to just show one example.

Line 333: A quick look at the IMPROVE data at the ROMO site in Rocky Mountain National Park also showed increased soil concentrations on 8/22, further corroborating the regional impact.

Line 348: Consider replacing "small" with "low". My first interpretation was with respect to particle size within the mode.

Line 367: While the hazy days corresponded to relatively high PM relative to non-hazy days, I am not sure this supports "large quantities". Removing "large quantities" would make a more defendable statement.

Line 381: Was the timing of the transport ever specifically discussed or provided?

Comments on Tables and Figures

Figure 4: Provide wavelength corresponding to AOD on this and subsequent figures.

Figures 7-13: See comment in text

Figure 14(a,b) and 15(a,b): Consider zooming in over North America.

Figure 18: Adding symbols would help with the error bars. As they are it is hard to tell which pair of upper and lower bars correspond to a single data point.

Figure 19: Add (a)-(e) in the caption. Consider changing "small PM2.5" and "large PM2.5" to "low" and "high".

---

## Referee Comment (RC2) · Anonymous Referee #2 · 26 May 2016

The manuscript presents an observation-based analysis of Colorado air quality impacted by long range transport of smoke particles from 2015 Pacific Northwest fires. Overall, the analysis is semi-qualitiatve at most; no transport modeling work is done, nor source-receptor relation is established with robustness. Synoptic chart and satellite data are used together to show the smoke transport pathways, but no new knowledge gained here. The manuscript argues that there is significant dust associated with smoke plume, but again, no figures to show where and when dust are uplifted. Can the dust be from great plains (such as west Nebraska) and not from fire region? it is a very interesting idea that biomass burning can uplift dust and such dust can transport with smoke plumes. The manuscript needs to show more quantiative supports for this

idea, either from analysis, modeling, or combined. Strong wind will uplift the soil dust, regardless. Specific concerns are listed below.

1. The manuscript's abstract and introduction gives readers an impression that the subject of the study is forest fires. But, in fact, in many cases, the fires studied here are fires in agcricultural areas (section 3.2). During the study time period, how much percentage of fires are from forest fires? This question is important because forest fires normally are bigger, inject smoke particles higher into the atmosphere for long range transport. Agricultural fires are smaller and don't injection smoke particles into the middle troposphere, but smoke particles from these fires can still transport in long distance and can be uplifted into the middle part of the atmosphere during the transport process. Together with the following papers, these points should be discussed either in the introduction or in the section 3.2 and 3.3.

Peterson, D., E. J. Hyer, and J. Wang , 2014: Quantifying the potential for high-altitude smoke injection in North American boreal forest using the standard MODIS fire products and sub-pixel-based methods, J. Geophys. Res. Atmos., 119, 3401-3419.

Colarco, P. R., M. R. Schoeberl, B. G. Doddridge, L. T. Marufu, O. Torres, and E. J. Welton, 2004: Transport of smokefrom Canadian forest fires to the surface near Washington, D.C.: Injection height, entrainment, and optical properties, J. Geophys. Res.,109, D06203, doi:10.1029/2003JD004248.

Wang, J., S. A. Christopher, U. S. Nair, J. S. Reid, E. M. Prins, J. Szykman, and J. L. Hand, 2006: Mesoscale modeling of Central American smoke transport to the United States: 1. "Top-down" assessment of emission strength and diurnal variation impacts, J. Geophys. Res., 111, D05S17, doi:10.1029/2005JD006416.

2. Line 46-47. Smoke particles not only affect clouds - so called indirect effect. They also have a semi-direct effect that affect cloud and atmospheric lapse rate through absorbing aerosols. In particular, when absorbing aerosols are above clouds, the semi-direct effect can be enhanced.

Ge, C., J. Wang , and J. S. Reid, 2014: Mesoscale modeling of smoke transport over the Southeast Asian Maritime Continent: coupling of smoke direct radiative feedbacks below and above the low-level clouds, Atmos. Chem. Phys. , 14, 159-174.

3. The paper used K and S as marker for biomass burning particles. However, it is good to use non-soil K instead of total K, as in Wang et al. (2006) and Kreidenweis et al. (2001). In addition, do biomass burning particles contain Ca, Al, etc.?

Kreidenweis, S. M., L. A. Remer, R. Bruintjes, and O. Dubovik, 2001: Smoke aerosols from biomass burning in Mexico: Hygroscopic smoke optical model, J. Geophys. Res., 106, 4831–4844.

4. Figure 18. how do you define relative mental mass concentrations? Relative to what? it should be in the figure caption. Figure 19 can be an interesting figure, but presenting the results in total amount for different specifies is confusing. More PM2.5 of course will have more chemical species. Relative percentage of these species with respect to total PM2.5 can be interesting to shown. In addition, any statistically significant test is conducted for panel a, -d. For example, in panel, there are significant variation of soil in small PM2.5 that can overlap with variation of soil in large PM2.5.

5. Line 314. "Dust and smoke from fires extended to 10 km". there is no evidence here that dust are from fires. Synoptic charts and back trajectory analysis show there is a high possibility that dust particles may from western part of Nebraska.

---

## Referee Comment (RC3) · Anonymous Referee #2 · 26 May 2016

in the review above, 'quantiative' and 'qualitiatve' are both typos. They should be 'quantitative'.

---

## Short Comment (SC1) · 20 Jun 2016

This paper is well written and presents a detailed case study of transported aerosols from wildfires in the Pacific Northwest contributing to poor air quality during August 2015 in the Colorado Front Range. However, the paper wrongly reports that concentrations of trace gases were not significantly impacted during this period. The study restricts its scope to the period of August 15 – September 2, during which three distinct smoke events occurred. This paper assumes the short periods of time between these events to be representative of normal or baseline conditions. But there is some evidence that the Front Range was still impacted by smoke during the intervening periods and that a comparison of periods prior to and following the smoke events may be more

appropriate for establishing changes to Front Range air quality. For example a comparison of the concentrations of CO at five CDPHE sites (CAMP, I25, Welby, Fort Collins Mason, and La Casa) between the three smoke events and several weeks before and after the period August 15 – September 2 shows significantly elevated CO during the smoke events. There are also changes in the abundances of O3 and NO2. We plan to submit an analysis of the impact of these smoke plumes on gas phase chemistry using 9 weeks of in situ data collected during summer 2015 at the Boulder Atmospheric Observatory (BAO) site. With this in mind, we suggest this paper restrict its focus to the aerosol impacts of the smoke events on Front Range air quality, and to simply remove the section on trace gas impacts. Additionally the broad phrase "air quality" in the title might be revised to "aerosol composition" or something similar to reflect this focus.

---

## Author Comment (AC1) · 2 Aug 2016

Thank you very much for pointing this out. We are happy to hear the Pacific Northwest fire influence on air quality along the Front Range is of interest to others and work is ongoing.

We have eliminated any discussion of gas phase species within our results (i.e., in the methods where we originally mentioned we did not see any changes in gas phase species measured by CDPHE sites during our study time period).

We agree that in light of this comment, a title revision is needed. Thus, we changed the title to, "Colorado air quality impacted by long-range transported aerosol: A set of

case studies during the 2015 Pacific Northwest fires" which still indicates the air quality was reduced in part due to aerosol from the fires.

---

## Author Comment (AC2) · 2 Aug 2016

*We would like to thank the reviewers for dedicating their time to provide constructive feedback on our manuscript, which has now improved from the original submission. We want to point out that line numbers listed below correspond to those in the revision and not the track changes version attached to this review.*

**Reviewer #1**

This is a well-written and interesting paper documenting the transport of wildfire smoke from the Pacific Northwest into Colorado. The paper is organized around specific events that resulted in degraded air quality and visibility through the Front Range of Colorado. Supporting evidence was provided through a variety of measurement platforms, including remote sensing and ground based measurements, as well as meteorological data and back trajectory analyses to describe the flow patterns during the events. The authors have combined these data into an interesting story that informs as to the transport of smoke across the United States with impacts on local air quality. An important result is the transport of mineral aerosols with the smoke plume. I recommend the paper be published after addressing the comments below.

*We thank the reviewer for his/her positive support and constructive review.*

Line 133: "g µm-3" typo.

*Typo fixed.*

Line 185: The authors repeatedly refer to "hazy" conditions along the Front Range (specifically Denver) and support the degraded air quality using PM data measured by the Colorado Department of Public Health and Environment. In checking the available data it appears that extinction data are also available from transmissometer measurements at the DESCI site. It would improve the paper to include these data so that the "hazy" can be quantified (line 185). In fact, the extinction values agree fairly closely with the TOPAZ lidar data in Figure 17 (given the wavelength differences).

*Thank you for pointing this out. We now include the DESCI site in Figure 1, include the DESCI beta extinction data in Figure 3, describe the site and extinction data in the methods (section 2.2), discuss the extinction data when indicating a time period is hazy (section 3.1), and compare it to TOPAZ (first part of section 3.4).*

Line 205: Please provide wavelength.

*We now provide the wavelength (550 nm) in the methods when MODIS is first introduced (line 87).*

Line 232, Section 3.3: This section is somewhat hard to follow because the figures are broken up so it requires flipping back and forth. I suggest organizing the figures so that, for example: the first event would include figure 7a-d, 8a, 9. It would reduce the number of figures and help to focus the discussion.

*We have revised so that each event corresponds to one figure as suggested by the reviewer. Now, Figures 7, 8, and 9 contain the RAP, HYSPLIT, and profiler data from Events 1, 2, and 3, respectively. We also made sure this change was reflected throughout the text.*

Line 244,245: Do the authors mean "northwesterly" here?

*Yes, this was fixed.*

Line 308: Figures 14-16 are similar enough it might be possible to just show one example.

*We agree that the figures are strikingly similar. As a result, we have stated that the observations of smoke and dust from CALIPSO was consistent for all event days in the text and added the other two event CALIPSO figures to a Supporting Information file.*

Line 333: A quick look at the IMPROVE data at the ROMO site in Rocky Mountain National Park also showed increased soil concentrations on 8/22, further corroborating the regional impact.

*Thank you for highlighting this. We also looked at fine mass, sulfur, and potassium concentrations, and those were also elevated on or near event days when IMPROVE samples were obtained (16, 22, 28 Aug). We evaluated the concentrations of these and soil on event versus non-event days in August, and noticed the concentrations for all were higher on the event days. We now discuss this in the text on lines 364-366 and added a figure showing the increased concentrations on event days as compared to non-event days in the Supporting Information.*

Line 348: Consider replacing "small" with "low". My first interpretation was with respect to particle size within the mode.

*Done.*

Line 367: While the hazy days corresponded to relatively high PM relative to non-hazy days, I am not sure this supports "large quantities". Removing "large quantities" would make a more defendable statement.

*Agreed, "large quantities" was removed.*

Line 381: Was the timing of the transport ever specifically discussed or provided?

*It was not originally, but we checked how far back the trajectories passed over the fire region (2 – 3 days). We now state this on line 407.*

Figure 4: Provide wavelength corresponding to AOD on this and subsequent figures.

*Done, but only for the first figure since the captions of the subsequent figures refer to the first.*

Figures 7-13: See comment in text

*Fixed.*

Figure 14(a,b) and 15(a,b): Consider zooming in over North America.

*We wanted to show that dust and smoke were indeed enhanced over the entire footprint of what CALIPSO observed for that transect (i.e., relative to a much larger scale). Thus, we did not zoom in on North America.*

Figure 18: Adding symbols would help with the error bars. As they are it is hard to tell which pair of upper and lower bars correspond to a single data point.

*Done. We also want to note that we restricted our XRF analysis from 27 Aug – 2 Sep due to the strange spike in concentrations on 26 Aug. After closer examination, we realized the data are likely not reliable on that day due to instrumental complications with temperature.*

Figure 19: Add (a)-(e) in the caption. Consider changing "small PM2.5" and "large PM2.5" to "low" and "high".

*Done.*

**Reviewer #2**

The manuscript presents an observation-based analysis of Colorado air quality impacted by long range transport of smoke particles from 2015 Pacific Northwest fires. Overall, the analysis is semi-quantitative at most; no transport modeling work is done, nor source-receptor relation is established with robustness. Synoptic chart and satellite data are used together to show the smoke transport pathways, but no new knowledge gained here. The manuscript argues that there is significant dust associated with smoke plume, but again, no figures to show where and when dust are uplifted. Can the dust be from great plains (such as west Nebraska) and not from fire region? it is a very interesting idea that biomass burning can uplift dust and such dust can transport with smoke plumes. The manuscript needs to show more quantitative supports for this idea, either from analysis, modeling, or combined. Strong wind will uplift the soil dust, regardless. Specific concerns are listed below.

*We understand the reviewer's concern, but disagree that more a quantitative analysis is needed. The focus is on the fact that dust is uplifted with the smoke and transported to the Front Range, where it was detected and impacted air quality. Directly showing the dust transport with the smoke is indeed a novel observation, particularly for this region as it has mostly been shown to occur in the dust belt region as we state in the introduction. Additionally we disagree that modeling is needed; we already provide ample evidence for our conclusions (i.e., multiple in situ measurements over the entire Front Range, remote sensing data from two different satellites and a lidar in Boulder, HYSPLIT air mass modeling analysis, in situ wind profiler data, and meteorological reanalysis fields).*

*We show direct evidence that the dust was transported with the smoke via CALIPSO and state that we evaluated CALIPSO in the surrounding regions to exclude trans-Pacific transport or other regional sources. We also use the meteorological data (e.g., the RAP and wind profiler analyses) and modeled HYSPLIT trajectories to support the sources of the air masses, which as the figures show, were likely not the Great Plains. However, we did include a more statistical HYSPLIT analysis to demonstrate that the fire plume regions indicated by MODIS data was indeed the major source (i.e., transport was dominant from these regions). See response to comment 5 for more details.*

1. The manuscript's abstract and introduction gives readers an impression that the subject of the study is forest fires. But, in fact, in many cases, the fires studied here are fires in agcricultural areas (section 3.2). During the study time period, how much percentage of fires are from forest fires? This question is important because forest fires normally are bigger, inject smoke particles higher into the atmosphere for long range transport. Agricultural fires are smaller and don't injection smoke particles into the middle troposphere, but smoke particles from these fires can still transport in long distance and can be uplifted into the middle part of the atmosphere during the transport process. Together with the following papers, these points should be discussed either in the introduction or in the section 3.2 and 3.3.

Peterson, D., E. J. Hyer, and J. Wang , 2014: Quantifying the potential for high-altitude smoke injection in North American boreal forest using the standard MODIS fire products and sub-pixel-based methods, J. Geophys. Res. Atmos., 119, 3401-3419.

Colarco, P. R., M. R. Schoeberl, B. G. Doddridge, L. T. Marufu, O. Torres, and E. J. Welton, 2004: Transport of smokefrom Canadian forest fires to the surface near Washington, D.C.: Injection height, entrainment, and optical properties, J. Geophys. Res.,109, D06203, doi:10.1029/2003JD004248.

Wang, J., S. A. Christopher, U. S. Nair, J. S. Reid, E. M. Prins, J. Szykman, and J. L. Hand, 2006: Mesoscale modeling of Central American smoke transport to the United States: 1. "Top-down" assessment of emission strength and diurnal variation impacts, J. Geophys. Res., 111, D05S17, doi:10.1029/2005JD006416.

*The images below show a map containing the most recent USGS land cover types (http://gis1.usgs.gov/csas/gap/viewer/land_cover/Map.aspx) in the left panel and the thermal anomalies (i.e., fire hotspots) detected by MODIS from the entire duration of the study overlaid on a shaded relief map (right panel). It is clear that most of the fires, with the exception of a few in north-central Washington, were located at high elevations and in forested areas. The fires in north-central Washington occurred on high elevation shrub and grassland, which we already state is a land type where fires were observed in the manuscript at the beginning of section 3.2. Almost no fires occurred on agricultural land, with the exception of a couple circled in the right panel. We incorporated this information into a supplementary figure (Figure S1) to support our statement in section 3.2.*

[Figure]

*Additionally, MODIS shows smoke transported from the fire hotspots, indicating an abundant fuel source to promote evolution of a dense smoke plume. For example, the image below contains MODIS corrected reflectance (true color) and thermal anomalies from 20 Aug. Smoke originating from these fires is prominent and travelled eastward towards the mountain states. The remaining days looked similar (see https://worldview.earthdata.nasa.gov/). Thus, the smoke was injected at altitudes where it could be transported long distances, and due to the apparent density of the smoke, were formed from sufficient fuel sources, such as forests.*

[Figure]

*Even though the fires we observed were predominantly from forested regions, and did indeed inject smoke plumes high enough into the atmosphere such that they were transported long distances, to encompass the fact that a couple of the fires detected during the study time period were from agricultural land, the first sentence of our introduction already stated, "Wildfires in both forested and agricultural regions serve as a steady source of pollutants into the atmosphere." We also now provide additional discussion and some of the references provided by the reviewer in the introduction on lines 41-44 regarding the fire size and injection height.*

2. Line 46-47. Smoke particles not only affect clouds - so called indirect effect. They also have a semi-direct effect that affect cloud and atmospheric lapse rate through absorbing aerosols. In particular, when absorbing aerosols are above clouds, the semidirect effect can be enhanced.

Ge, C., J. Wang , and J. S. Reid, 2014: Mesoscale modeling of smoke transport over the Southeast Asian Maritime Continent: coupling of smoke direct radiative feedbacks below and above the low-level clouds, Atmos. Chem. Phys. , 14, 159-174.

*Thank you for bringing this to our attention. We now discuss this and added the reference on lines 52-53.*

3. The paper used K and S as marker for biomass burning particles. However, it is good to use non-soil K instead of total K, as in Wang et al. (2006) and Kreidenweis et al. (2001). In addition, do biomass burning particles contain Ca, Al, etc.?

Kreidenweis, S. M., L. A. Remer, R. Bruintjes, and O. Dubovik, 2001: Smoke aerosols from biomass burning in Mexico: Hygroscopic smoke optical model, J. Geophys. Res., 106, 4831–4844.

*We calculated non-soil K and soil K and have included these in the new Supporting Information file. We also highlight in the text on lines 361-364 that both non-soil and soil K concentrations were higher during the event time period.*

*Biomass burning aerosols have been shown to contain metals such as Mg, Al, Ca, Cr, Mn, Fe, Ni, Cu, Zn, but it has been suggested that the biomass may have accumulated metal-containing species that were re-emitted during biomass burning, thus the metals may have originated from other sources, such as dust. Although, the exact sources of these metals in biomass burning aerosols remains unknown. Further, if leached from the ground, it is probable that the concentrations of these metals in biomass burning is negligible compared to those in mineral or soil dust. We now explain this and include the following reference on lines 373-376.*

*Chang-Graham, A. L., Profeta, L. T. M., Johnson, T. J., Yokelson, R. J., Laskin, A., and Laskin, J.: Case Study of Water-Soluble Metal Containing Organic Constituents of Biomass Burning Aerosol, Environ Sci Technol, 45, 1257-1263, 2011.*

4. Figure 18. how do you define relative mental mass concentrations? Relative to what? it should be in the figure caption. Figure 19 can be an interesting figure, but presenting the results in total amount for different specifies is confusing. More PM2.5 of course will have more chemical species. Relative percentage of these species with respect to total PM2.5 can be interesting to shown. In addition, any statistically significant test is conducted for panel a, -d. For example, in panel, there are significant variation of soil in small PM2.5 that can overlap with variation of soil in large PM2.5.

*These are relative to the maximum concentration measured from each species, which we now state in the caption of (now) Figure 12. We did this to enable the increases during influences from the fires to be apparent in all the metals, otherwise metals with generally low concentrations (i.e., K) would be buried near zero relative to metals that are generally higher in concentration (i.e., Si). By showing the relative metal mass in this way, it is clear that each metal we discuss is higher in concentration during fire influences as compared to days with a lesser or no influence from the fires.*

*It is not necessarily true that more $PM_{2.5}$ will have more of each chemical species; take As and Pb shown in (now) Figure 13, for example. Those metals are lower in concentration when $PM_{2.5}$ is higher. We conducted a statistical significance test for SOIL and $PM_{2.5}$ (t-test: two sample assuming unequal variances) and the differences were statistically significant (t-Stat = 2.23 and t-Critical = 1.67). The metal concentration averages in the other panels were also statistically significant according to the t-test. We now note this in the caption.*

5. Line 314. "Dust and smoke from fires extended to 10 km". there is no evidence here that dust are from fires. Synoptic charts and back trajectory analysis show there is a high possibility that dust particles may from western part of Nebraska.

*It is evident by the back trajectory analysis that air masses did not travel over Nebraska nor the Great Plains on the worst event days shown in red (see e panels in revised Figures 7, 8, and 9). On occasion, surrounding days did pass over the Great Plains (blue dashed lines), but occurred 5 to 10 days back and prior to passing over the fire plume regions (see MODIS data in Figures 4, 5 ,and 6). Only 5 of the 48 trajectories passed over the Great Plains during Event 1, none during Event 2, and 3 of the 48 during Event 3. Thus, the likelihood that the Great Plains played a major source relative to the region where the fire plumes were located is unlikely based on HYSPLIT statistics (see table below), which we now discuss in section 3.3. We also now point out that Event 2, which was the worst in terms of $PM_{2.5}$ and total-column extinction (Figure 3), also had the most transport from the fire regions. The same relationship holds true to remaining events, i.e., the highest (lowest) % transport, the higher (lower) the $PM_{2.5}$ and extinction.*

| Event | Date | Total # of trajectories | # of trajectories that passed through fire plumes | % of trajectories that passed through fire plumes | per event | # of trajectories that passed through Plains | % of trajectories that passed through Plains | per event |
|---|---|---|---|---|---|---|---|---|
| 1 | 15-Aug | 12 | 1 | 8% | 40% | 3 | 25% | 10% |
| | 16-Aug | 12 | 3 | 25% | | 2 | 17% | |
| | **17-Aug** | **12** | **6** | **50%** | | **0** | **0%** | |
| | 18-Aug | 12 | 9 | 75% | | 0 | 0% | |
| 2 | 20-Aug | 12 | 12 | 100% | 96% | 0 | 0% | 0% |
| | 21-Aug | 12 | 12 | 100% | | 0 | 0% | |
| | 22-Aug | 12 | 11 | 92% | | 0 | 0% | |
| | **23-Aug** | **12** | **11** | **92%** | | **0** | **0%** | |
| 3 | 26-Aug | 12 | 11 | 92% | 85% | 3 | 25% | 6% |
| | 27-Aug | 12 | 8 | 67% | | 0 | 0% | |
| | 28-Aug | 12 | 10 | 83% | | 0 | 0% | |
| | **29-Aug** | **12** | **12** | **100%** | | **0** | **0%** | |

*Additionally, 500 hPa geopotential heights (see a and b panels in Figures 7, 8, and 9) clearly show westerly to northwesterly flow along much of the western U.S. and Colorado, and even in Nebraska, indicating transport from those directions and not from Nebraska. Based on this evidence from modelling and reanalysis, a "high probability" of dust arriving from Nebraska is not likely.*

[revised manuscript text omitted]

Figure S1. (a) Land cover data from the U.S. Geological Survey determined using multi-season satellite imagery from 1999–2001 in conjunction with digital elevation model (DEM) derived datasets (e.g. elevation, landform) to model natural and semi-natural vegetation (http://gis1.usgs.gov/csas/gap/viewer/land_cover/Map.aspx). (b) Elevation data from the U.S. Forest Service (http://apps.fs.fed.us/fiadb-downloads/CSV/datamart_csv.html; surveys vary by state, but were conducted 1989–2014) in addition to all MODIS thermal anomalies (i.e., fire hotspots) detected during the study time period. Fires occurred on higher elevation forest, shrub, and grass lands, but predominantly in forested or woodland areas. Few fires occurred on lower elevation agricultural lands.

[Figure]

Figure S2. Same as Figure 10 in the manuscript, but for the night prior to Event 2 and from 22 Aug 2015 09:19:24 UTC.

[Figure]

Figure S3. Same as Figure 10 in the manuscript, but for the day of Event 3 and from 29 Aug 2015 09:24:15 UTC.

[Figure]

Figure S4. Concentrations of total PM$_{2.5}$, soil, S, and K from the IMPROVE monitoring site in Rocky Mountain National Park (ROMO; 40.28°N, 105.55°W; 2,760 m MSL) during event days and non-event days during the month of August 2015. Data and information on sampling and analytical protocols are found at http://views.cira.colostate.edu/fed/ (Malm et al., 1994; Hand et al., 2011). Eleven samples were collected and analyzed during this time period. Uncertainty values are provided by IMPROVE for total mass and element concentration measurements (S and K), but not soil.

[Figure]

Figure S5. Non-soil and soil K concentrations measured by the PX-375 during the study time period. Concentrations were calculated from total K based on the methods on Kreidenweis et al. (2001), where [non-soil K] = [K] – 0.6[Fe] and [soil K] = [total K] – [non-soil K]. It is apparent that non-soil K and soil K were present during the Event 3 time period when the PX-375 was operable. Time periods without measurements were due to the K concentrations being below the LDL.